# Neutrophil-mediated fibroblast-tumor cell il-6/stat-3 signaling underlies the association between neutrophil-to-lymphocyte ratio dynamics and chemotherapy response in localized pancreatic cancer: A hybrid clinical-preclinical study

Iago de Castro Silva[1], Anna Bianchi[1], Nilesh U Deshpande[1], Prateek Sharma[1,2], Siddharth Mehra[1], Vanessa Tonin Garrido[1], Shannon Jacqueline Saigh[3], Jonathan England[4], Peter Joel Hosein[3,5], Deukwoo Kwon[6], Nipun B Merchant[1,6], Jashodeep Datta[1,6]*

[1]Department of Surgery, University of Miami Miller School of Medicine, Miami, United States; [2]Department of Surgery, University of Nebraska Medical Center, Omaha, United States; [3]Sylvester Comprehensive Cancer Center, Miami, United States; [4]Department of Pathology, University of Miami, Miami, United States; [5]Department of Medicine, University of Miami, Miami, United States; [6]Department of Public Health Sciences, The University of Texas Health Science Center at Houston, Houston, United States

*For correspondence:
jash.datta@med.miami.edu

Competing interest: The authors declare that no competing interests exist.

## Abstract

**Background:** Partial/complete pathologic response following neoadjuvant chemotherapy (NAC) in pancreatic cancer (PDAC) patients undergoing pancreatectomy is associated with improved survival. We sought to determine whether neutrophil-to-lymphocyte ratio (NLR) dynamics predict pathologic response following chemotherapy in PDAC, and if manipulating NLR impacts chemosensitivity in preclinical models and uncovers potential mechanistic underpinnings underlying these effects.

**Methods:** Pathologic response in PDAC patients (n=94) undergoing NAC and pancreatectomy (7/2015-12/2019) was dichotomized as partial/complete or poor/absent. Bootstrap-validated multivariable models assessed associations between pre-chemotherapy NLR (%neutrophils÷%lymphocytes) or NLR dynamics during chemotherapy (ΔNLR = pre-surgery—pre-chemotherapy NLR) and pathologic response, disease-free survival (DFS), and overall survival (OS). To preclinically model effects of NLR attenuation on chemosensitivity, $Ptf1a^{Cre/+}$; $Kras^{LSL-G12D/+}$;$Tgfbr2^{flox/flox}$ (PKT) mice and C57BL/6 mice orthotopically injected with $Kras^{LSL-G12D/+}$;$Trp53^{LSL-R172H/+}$;$Pdx1^{Cre}$(KPC) cells were randomized to vehicle, gemcitabine/paclitaxel alone, and NLR-attenuating anti-Ly6G with/without gemcitabine/paclitaxel treatment.

**Results:** In 94 PDAC patients undergoing NAC (median:4 months), pre-chemotherapy NLR (p<0.001) and ΔNLR attenuation during NAC (p=0.002) were independently associated with partial/complete pathologic response. An NLR score = pre-chemotherapy NLR+ΔNLR correlated with DFS (p=0.006) and OS (p=0.002). Upon preclinical modeling, combining NLR-attenuating anti-Ly6G treatment with gemcitabine/paclitaxel—compared with gemcitabine/paclitaxel or anti-Ly6G alone—not

only significantly reduced tumor burden and metastatic outgrowth, but also augmented tumor-infiltrating CD107a[+]-degranulating CD8[+] T-cells (p<0.01) while dampening inflammatory cancer-associated fibroblast (CAF) polarization (p=0.006) and chemoresistant IL-6/STAT-3 signaling in vivo. Neutrophil-derived IL-1β emerged as a novel mediator of stromal inflammation, inducing inflammatory CAF polarization and CAF-tumor cell IL-6/STAT-3 signaling in ex vivo co-cultures.

**Conclusions:** Therapeutic strategies to mitigate neutrophil-CAF-tumor cell IL-1β/IL-6/STAT-3 signaling during NAC may improve pathologic responses and/or survival in PDAC.

**Funding:** Supported by KL2 career development grant by Miami CTSI under NIH Award UL1TR002736, Stanley Glaser Foundation, American College of Surgeons Franklin Martin Career Development Award, and Association for Academic Surgery Joel J. Roslyn Faculty Award (to J. Datta); NIH R01 CA161976 (to N.B. Merchant); and NCI/NIH Award P30CA240139 (to J. Datta and N.B. Merchant).

## Editor's evaluation

Iago De Castro et al. is a fundamental new study that conveys to readers that neutrophil-to-lymphocyte ratio dynamics could predict pancreatic cancer pathologic response to neoadjuvant therapy. The study is compelling in that it specifically provides means to determine the effect that front-line neoadjuvant therapy could have on the function of key microenvironmental cells (e.g., T Cell and Cancer-associated fibroblast) if combined with an anti-Ly6G treatment.

## Introduction

Modern multi-agent chemotherapy delivered in the neoadjuvant setting for localized pancreatic ductal adenocarcinoma (PDAC) is an increasingly popular treatment sequencing strategy (**Datta and Merchant, 2021**). Our group has previously reported that major pathologic response following neoadjuvant chemotherapy (NAC) is associated with improved overall survival (**Macedo et al., 2019**). A major unmet need that remains is the discovery of biomarkers of pathologic response as well as subsequent disease trajectories in patients who undergo resection following NAC.

Neutrophil-to-lymphocyte ratio (NLR) has emerged as a promising biomarker in localized and advanced PDAC. Beyond its prognostic value in advanced unresectable disease (**Iwai et al., 2020**), recent evidence implicates the value of pre-surgery NLR in forecasting recurrence in patients undergoing upfront pancreatectomy (**Nywening et al., 2018**), as well as pre- and post-treatment NLR in predicting pathologic response following neoadjuvant chemoradiotherapy (**Hasegawa et al., 2016**; **Kubo et al., 2019**). However, the precise relationship between NLR *dynamics during* neoadjuvant treatment and pathologic response and/or survival in localized PDAC patients undergoing pancreatectomy has not been previously explored.

Emerging evidence implicates stromal inflammation in the PDAC tumor microenvironment (TME)—predominantly through inflammatory polarization of cancer-associated fibroblasts (iCAF) and CAF-derived secretion of IL-6 (**Öhlund et al., 2017**)—as a major driver of chemoresistance in PDAC (**Hosein et al., 2020**). Furthermore, prior work from our group has revealed that CAF-derived IL-6 engages in tumor-permissive crosstalk by activating STAT3 signaling within tumor cells (**Nagathihalli et al., 2016**), and that heightened CAF-tumor cell IL-6/STAT-3 signaling crosstalk is a central mediator of chemoresistance in PDAC (**Dosch et al., 2021**). As such, how tumor-permissive inflammatory cues such as neutrophil-lymphocyte balance intersect with such signaling mechanisms underlying therapeutic resistance in PDAC remains critically underexplored.

In a cohort of patients with operable PDAC undergoing modern multi-agent NAC, we sought to determine if *NLR dynamics* predict pathologic response following NAC in patients undergoing curative-intent pancreatectomy. We further investigated if pharmacologically modulating NLR dynamics in preclinical models of PDAC would impact chemosensitivity and uncover potential immunologic- and stromal-mediated mechanisms underlying these effects in vivo.

# Materials and methods

**Key resources table**

| Reagent type (species) or resource | Designation | Source or reference | Identifiers | Additional information |
|---|---|---|---|---|
| Cell line (*Mus musculus*) | Pancreatic Tumor Cells from $Kras^{LSL-12D/+}$;$Trp53^{R172H/+}$; $Pdx1^{Cre}$ (KPC) mouse | Ben Stanger/UPenn | KPC6694c2 | |
| Cell line (*Mus musculus*) | Tumor associated fibroblasts from KPC mouse | ***Nagathihalli et al., 2016*** | KPC CAFs | |
| Other | $Ptf1a^{Cre/+}$;$Kras^{LSL-G12D/+}$;$Tgfbr2^{flox/flox}$ | ***Datta et al., 2022*** | PKT | Genetically engineered mouse |
| Antibody | Anti-Ly6G (Rat monoclonal) reactive to mouse | BioXcell | Clone 1A8 Catalog# BE0075-1 | 25 µg/dose |
| Antibody | Anti-IL-1β neutralizing antibody (*E. coli*, polyclonal) | R&D Systems | Catalog# AF-401-NA | 1:80 |
| Antibody | Cxcl1 (Rabbit, monoclonal) Reactive to human and mouse | Abcam | Catalog# ab86436 | 1:500 |
| Antibody | Podoplanin (Mouse, monoclonal) Reactive to human | Cell Signalling | Catalog# 26981 | 1:200 |
| Antibody | Podoplanin (Syrian hamster, monoclonal) Reactive to mouse | Abcam | Catalog# ab92319 | 1:200 |
| Antibody | CD3 (170Er, Human, monoclonal) 3170019D | Fluidigm | 3170019D | 1:1000 |
| Antibody | CD11B (149Sm, Human, monoclonal) | Fluidigm | 3149028D | 1:1000 |
| Antibody | α-SMA (141Pr, Human, monoclonal) | Fluidigm | 314017D | 1:1000 |
| Antibody | Pan-Cytokeratin (148Nd, Human, monoclonal) | Fluidigm | 3148022D | 1:1000 |
| Antibody | CD15 (164Dy, Human, monoclonal) | Fluidigm | 3164001B | 1:1000 |
| Antibody | CD8 (146Nd, Human, monoclonal) | Fluidigm | 3146001B | 1:1000 |
| Chemical compound, drug | Anakinra | SOBI Pharmaceuticals | α-IL-1R1 inhibitor | |
| Sequence-based reagent | *Cxcl1* Primer - Mouse | Qiagen | Gene ID - QT00115647 | |
| Sequence-based reagent | *Il6* Primer - Mouse | Qiagen | Gene ID - QT00098875 | |
| Commercial assay or kit | Cytokine array - Mouse | R&D Systems | ARY006 | |

## Clinical analysis

Patients with localized PDAC who received NAC with either mFOLFIRINOX, gemcitabine/abraxane, or both and underwent pancreatectomy between July 2015 and December 2019 at a tertiary academic center (n=101) were enrolled. Patients were excluded if annotated pathologic response information was unavailable (n=5) or if they underwent R2 resection (n=2; *Figure 1A*; *Appendix 1—table 1*). Pathologic response (PR) in resected specimens was dichotomized as 'partial/complete' or 'poor/absent' response based on established College of American Pathologists guidelines (***Washington et al., 2010***). For each patient, the proportion of neutrophils and lymphocytes were obtained from complete blood counts accrued at two timepoints—pre-NAC and pre-surgery (for details, see **Appendix**). NLR was defined as %neutrophils÷%lymphocytes. Both the absolute NLR prior to initiation of NAC (pre-NAC aNLR) as well as dynamic changes in NLR *during* NAC, defined as ΔNLR (=pre-surgery aNLR *minus* pre-NAC aNLR; *Park et al., 2020*), were correlated with PR. Multivariable models assessed the independent association of aNLR and ΔNLR metrics (dichotomized into high vs. low) with partial/complete PR. Area under receiver-operating curves (AUCs) were estimated for three models (aNLR only, ΔNLR only, combined aNLR+ΔNLR), internally validated using bootstrap logistic regression, and an 'NLR score' comprising product of regression coefficients and aNLR/ΔNLR ( = 10.45–2.9224*aNLR - 2.13*ΔNLR) was generated to stratify disease-free (DFS) and overall survival (OS) via Kaplan-Meier estimates. All tests were two-sided and statistical significance designated as p≤0.05.

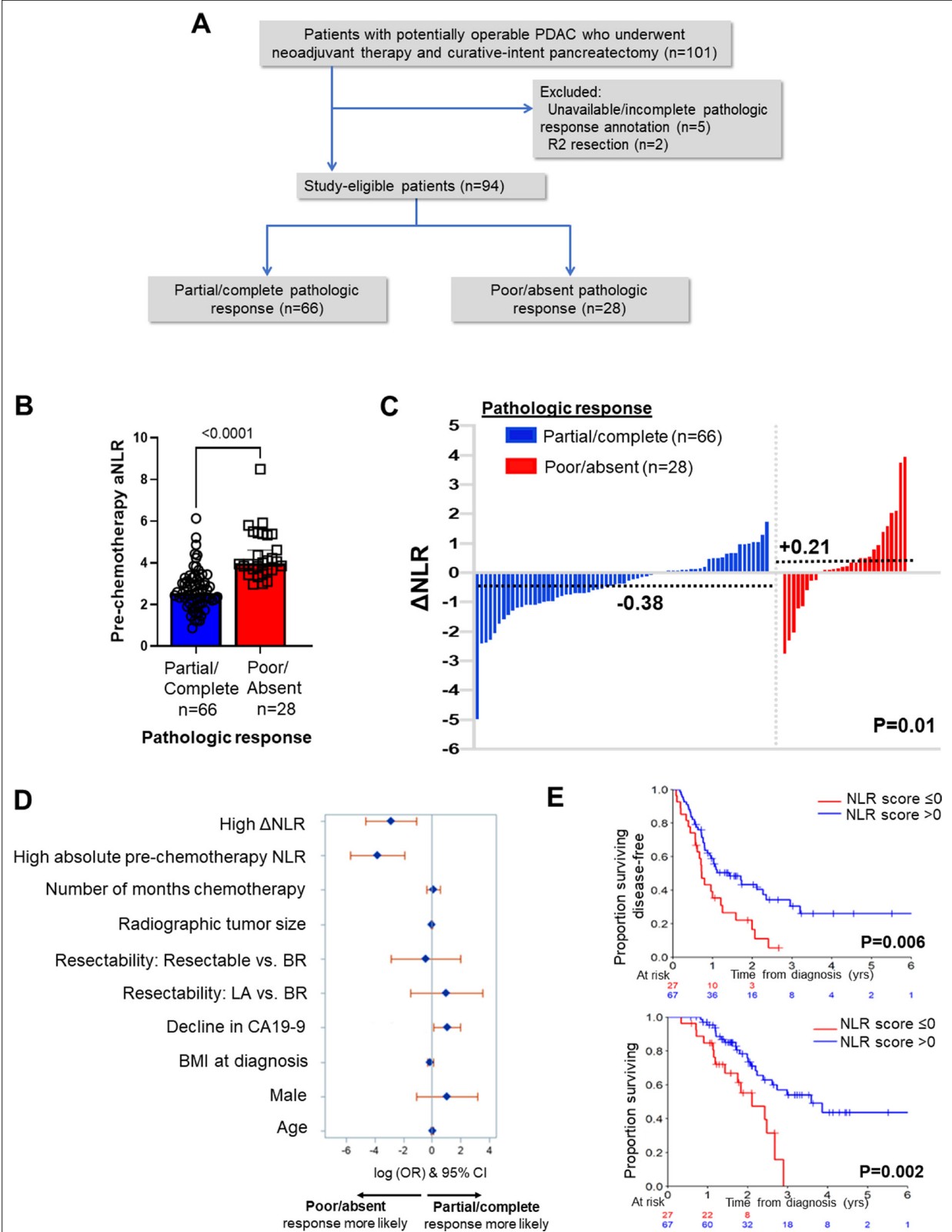

**Figure 1.** Neutrophil-to-lymphocyte ratio (NLR) dynamics are associated with pathologic response and survival following neoadjuvant chemotherapy in pancreatic cancer. (**A**) STROBE diagram for selection of study-eligible patients with potentially operable pancreatic ductal adenocarcinoma undergoing neoadjuvant chemotherapy and curative-intent pancreatectomy, stratified by pathologic response; (**B**) Comparison of pre-chemotherapy absolute NLR (aNLR) between resected PDAC patients who demonstrated partial/complete pathologic response (n=66) and poor/absent pathologic response

*Figure 1 continued on next page*

*Figure 1 continued*

(n=28) following neoadjuvant chemotherapy. Median (IQR) values are plotted; (**C**) Waterfall plot depicting the delta-NLR (ΔNLR = pre-surgery NLR—pre-chemotherapy NLR) of all study-eligible patients, stratified by partial/complete (*blue*) or poor/absent (*red*) pathologic response. Dotted lines indicate median ΔNLR in each cohort, and adjoining p-value represents the comparison of these median values; (**D**) Forest plot showing predictors of pathologic response following NAC in a multivariable logistic regression model. Adjusted log odds ratios (ORs) and corresponding 95% confidence intervals are plotted on the x-axis; (**E**) Stratification of disease-free survival (*top*) and overall survival (*bottom*) by 'NLR score', calculated as the product of regression coefficients and aNLR/ΔNLR. The NLR score was dichotomized at ≤0 or>0 based on its efficiency at prognosticating DFS and OS. Number of patients at risk at each time point shown in adjoining tables.

## In vivo experiments

To recapitulate systemic NLR attenuation in preclinical models of PDAC, C57BL/6 mice orthotopically injected with $50 \times 10^3$ syngeneic $Kras^{LSL-G12D/+};Trp53^{LSL-R172H/+};Pdx1^{Cre}$ PDAC cells (KPC6694c2, provided by Ben Stanger/UPenn, mycoplasma negative) were treated with increasing doses of neutralizing anti-Ly6G antibody (BioXcell; 25 μg, 100 μg, 200 μg) to attenuate—but not deplete—circulating $Ly6G^+:CD3^+$ ratios for further experiments.

C57BL/6 mice were then orthotopically injected with $50 \times 10^3$ KPC6694c2 (henceforth KPC) cells and randomized into four groups starting 10 days after tumor inoculation (n=8–10/arm): vehicle control, NLR-attenuating anti-Ly6G alone (25 μg/dose) q3 days starting day 10, gemcitabine (100 mg/kg) and paclitaxel (10 mg/kg) once weekly starting day 14, and gemcitabine/paclitaxel treatment (day 14) following a 'priming' phase of anti-Ly6G attenuation starting day 10 (for details, see **Appendix**). Mice were sacrificed following 3 weeks of treatment, tumor burden and metastatic outgrowth evaluated, and tumor samples subjected to histological analysis, immunohistochemistry (Ly6G/Gr1, phosphorylated STAT3, cleaved caspase-1, and CD31), flow cytometric CAF and immune phenotyping, and enzyme-linked immunosorbent assay (ELISA; IL-6 and IL-1β; *Park et al., 2020*). A similar series of experiments were performed in $Ptf1a^{Cre/+};Kras^{LSL-G12D/+};Tgfbr2^{flox/flox}$ (PKT) genetically engineered mice (*Datta et al., 2022*) to validate observations from the orthotopic KPC model (for details, see **Appendix**).

## Imaging mass cytometry (IMC) in human PDAC tumors

We retrieved FFPE blocks of 6 pre-treatment PDAC specimens from localized PDAC patients who underwent neoadjuvant chemotherapy and surgical resection, and stratified these post-hoc into partial/complete (n=3) or poor/absent (n=3) pathologic response. For detailed clinical annotation of these specimens, see *Appendix 1—table 2*. A board-certified GI pathologist selected regions of interest (ROI) from each slide comprising tumor cells, fibroblasts, and immune cells by correlating with corresponding H&E-stained sections. This slide was stained with an IMC panel of 10 metal-conjugated antibodies and a cell intercalator (**Appendix**). Prior to acquisition, Hyperion mass cytometry system (Fluidigm) was autotuned using a 3-element tuning slide and detection threshold of >700 mean duals of 175Lu was used according to manufacturer protocol. ROIs (1.8–3 mm$^2$) were ablated and acquired at 200 Hz. Data were exported as MCD files and analyzed for single-cell segmentation analysis using Visiopharm software. For details, refer to **Appendix**.

## Ex vivo co-culture experiments

Tumor-infiltrating $Ly6G^+F4/80^-$ neutrophils from orthotopic KPC tumor-bearing mice were isolated from fresh tumor suspensions using the Myeloid Derived Suppressor Cell Isolation Kit and QuadroMACS Separator (Miltenyi Biotech), and: (1) subjected to multiplex cytokine arrays using Proteome Profiler Mouse Cytokine Array Kit (R&D Systems, Minneapolis, MN); and (2) co-cultured with KPC CAFs with or without concurrent treatment with anti-IL-1β neutralizing antibody (Thermofisher, Waltham, MA) and IL-1R1 inhibitor Anakinra (SOBI Pharmaceuticals, Sweden). KPC tumor cells were incubated with conditioned media harvested from ex vivo co-cultures of intratumoral neutrophils and CAFs, either alone or with anti-IL-1β or anti-IL-6 neutralizing antibodies (Thermofisher, Waltham, MA), and ensuing protein lysates blotted for phosphorylated STAT-3 (pSTAT3). For complete details of all preclinical, in vivo, and in vitro experiments, see **Appendix**.

## Results

### NLR dynamics during NAC as biomarker of pathologic response and survival

Of 94 eligible patients (mean age 67, 58% female, 6% with germline homologous recombination deficiency genotype [*BRCA2,* n=5; *PALB2,* n=1]), 78% had borderline resectable or locally advanced disease. Patients received a median of 4 months of NAC (range 2–14), 52% received mFOLFIRINOX, and a minority of patients (6%) received neoadjuvant radiotherapy. Following NAC, partial/complete PR was achieved in 70% (66/94) while 28 patients (30%) demonstrated poor/absent PR (*Appendix 1—table 1*). Median pre-NAC aNLR was significantly lower in patients with partial/complete PR compared with poor/absent PR (2.53 vs 3.97; p<0.001) (*Figure 1B*). Moreover, a net *attenuation* in ΔNLR was observed in patients demonstrating partial/complete PR compared with a net *increase* in ΔNLR in poor/absent PR (median –0.38 vs. +0.21; p=0.01 respectively; *Figure 1C*). Of note, median aNLR or ΔNLR did not differ significantly between patients who received neoadjuvant mFOLFIRINOX vs. gemcitabine/*nab*-paclitaxel (3.00 vs 3.09, p=0.47; –0.02 to –0.25, p=0.16).

On multivariable modeling, higher aNLR (OR 0.02, 95% CI 0.003–0.15; p<0.001 [Ref: low aNLR]), higher ΔNLR (OR 0.06, 95% CI 0.01–0.33; p=0.002 [Ref: low ΔNLR]), and any %decline in CA19-9 during NAC (OR 1.82, 95% CI 0.001–3.74; p=0.05 [Ref: any %increase in CA19-9])—but not NAC duration, BMI, resectability status, or use of neoadjuvant radiotherapy—were independent predictors of achieving partial/complete PR following NAC (*Figure 1D*, *Appendix 1—table 3*). Bootstrap-validated AUC-derived analysis revealed that a combined NLR model encompassing both aNLR and ΔNLR most efficiently predicted PR with an AUC of 0.96 (*Appendix 1—figure 1*). This combined NLR model also effectively estimated bootstrap-validated time-dependent AUC for both DFS (2 year: 0.61, 95% CI 0.56–0.65) and OS (2 year: 0.60, 95% CI 0.55–0.67) in this cohort (*Appendix 1—figure 2A–C*).

At a median follow-up of 30 (IQR 7–49) months, 2 year and 5 year survival in this selected cohort of patients undergoing resection were 59% and 34%, respectively. An NLR score comprising the product of regression coefficients and aNLR/ΔNLR dichotomized at <0 and≥0 provided strongest discrimination of DFS and OS. Patients with an NLR score ≤0 demonstrated improved DFS (median 1.4 vs 0.7 years; p=0.006) and OS (median 3.6 vs 2.1 years; p=0.002) compared with patients with an NLR score >0 (*Figure 1E*).

### Attenuation of NLR potentiates chemosensitivity in murine PDAC

To model NLR attenuation in preclinical models of PDAC, treatment of orthotopic KPC tumor-bearing mice with neutrophil-attenuating 25 μg anti-Ly6G dosing achieved approximately 50% attenuation in circulating Ly6G$^+$:CD3$^+$ NLR ratio compared with vehicle treatment (*Appendix 1—figure 3A*); treatment with gemcitabine/paclitaxel, however, did not significantly decrease Ly6G$^+$:CD3$^+$ ratios (*Appendix 1—figure 3B*). While gemcitabine/paclitaxel treatment expectedly decreased PDAC tumor size compared with vehicle and anti-Ly6G alone treatments, concurrent treatment of tumor-bearing mice with gemcitabine/paclitaxel +anti-Ly6G further significantly decreased pancreatic tumor weight (*Figure 2A*) and metastatic outgrowth, graded by the presence of tumor deposits at six extra-pancreatic sites (*Figure 2B*). Importantly, mice treated with combination gemcitabine/paclitaxel +anti-Ly6G treatment did not incur additional systemic toxicity during treatment as measured by mouse weights (*Appendix 1—figure 4A*) and systemic ALT levels (*Appendix 1—figure 4B*).

Compared with gemcitabine/paclitaxel alone, NLR attenuation with anti-Ly6G improved chemosensitivity as evidenced by significantly decreased tumor area by H&E staining (p=0.01; *Figure 2C*) as well as increased cleaved caspase-3 and microvessel (CD31) density (*Figure 2D*) in gemcitabine/paclitaxel +anti-Ly6G treated mice compared with all other treatment groups.

To validate these observations in a spontaneous PDAC mouse model, we treated 4-week-old PKT mice with vehicle, gemcitabine/paclitaxel alone, and gemcitabine/paclitaxel plus anti-Ly6G combinations for 2 weeks. In this model as well, NLR attenuation with anti-Ly6G improved chemosensitivity vs. chemotherapy alone as evidenced by decreased primary tumor weights (p=004; *Figure 2E*) and tumor area by H&E staining (P=0.008; *Figure 2F*) at endpoint analysis.

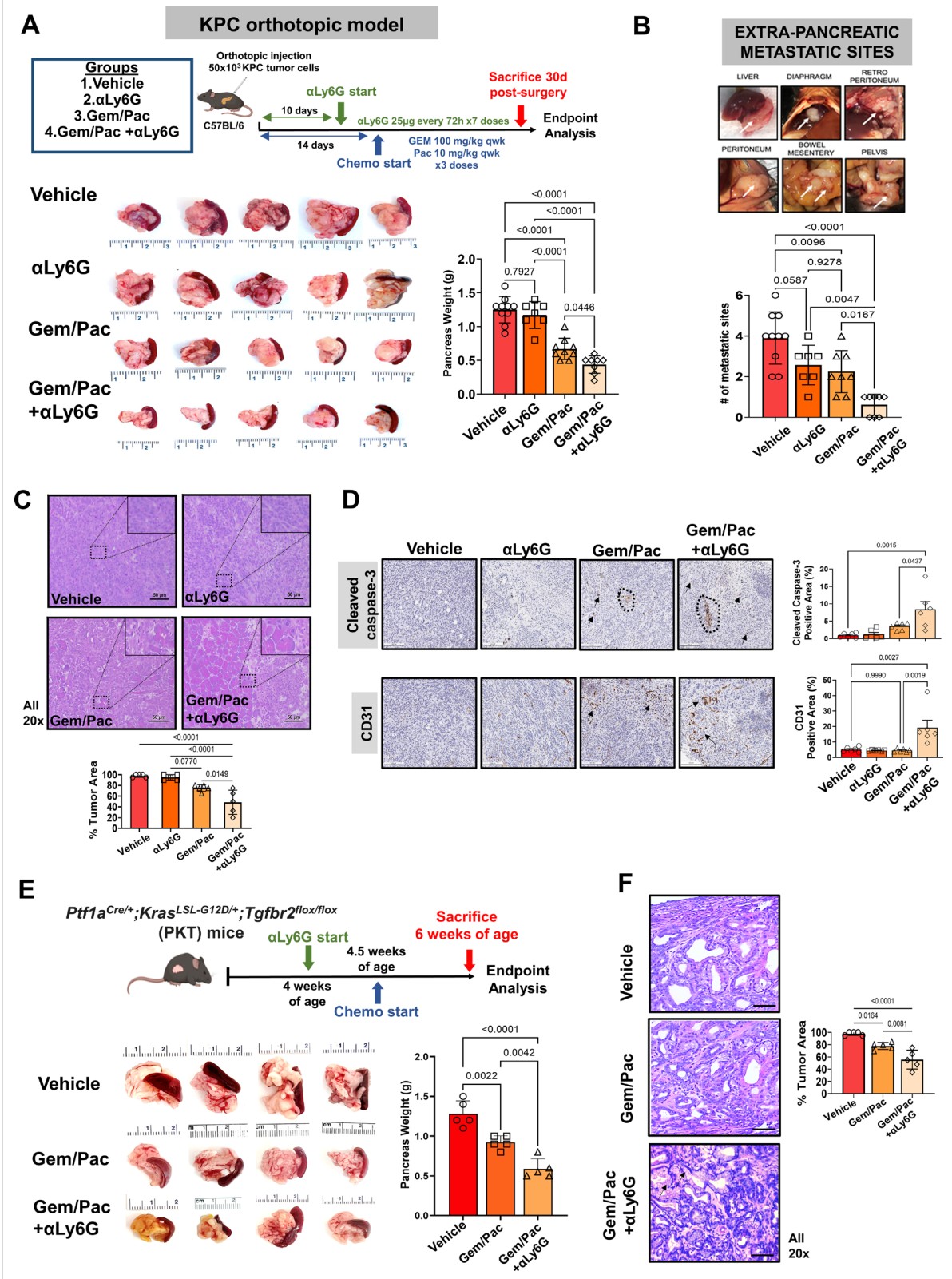

**Figure 2.** Attenuating neutrophil-to-lymphocyte ratio (NLR) improves sensitivity to chemotherapy in preclinical models of pancreatic cancer. (**A**) Schematic of in vivo experimental design, illustrating treatment groups utilized (vehicle, anti-Ly6G [αLy6G] alone, gemcitabine/paclitaxel alone, and gemcitabine/paclitaxel+αLy6G), treatment timing, and schedules/regimens in KPC orthotopic model (*top*). Representative images (n=5 biologic replicates) from primary pancreatic tumors at endpoint analysis in each treatment group and adjoining histogram demonstrating differences in whole

*Figure 2 continued on next page*

*Figure 2 continued*

pancreas weights between treatment groups (n=8–10 mice/arm) at sacrifice are depicted (*bottom*); (**B**) Metastatic outgrowth in KPC orthotopic models of PDAC is graded by presence of tumor deposits at six extra-pancreatic sites; a representative example from a vehicle-treated mouse in these experiments is shown. Adjoining histogram depicts comparison of the frequency of extra-pancreatic metastatic involvement (values 1 through 6 for each mouse) across treatment groups (n=8–10 mice/group); (**C**) Representative images of tumor sections from each treatment group stained by H&E to demonstrate tumor area (all 20 x; scale bar = 50 μm), with high-magnification insets (40 x) indicating relevant areas on these representative sections. Slides from each treatment group were blinded, and %tumor area quantified by a board-certified pathologist (n=5 from each treatment group). This comparison is depicted in adjoining histogram; (**D**) Representative images of tumor sections stained for cleaved caspase-3 (CC-3) and CD31 from each treatment group (n=5; all 20 x; scale bar = 200 μm). Dotted circles and arrows represent areas of positive staining. Adjoining histograms show quantification of cleaved caspase-3 and CD31 staining across treatment groups (n=5 mice/group); (**E**) Representative images from primary pancreatic tumors at endpoint analysis in indicated treatment groups in the *Ptf1a*$^{Cre/+}$;*Kras*$^{LS-L-G12D/+}$;*Tgfbr2*$^{flox/flox}$ (PKT) genetically engineered mouse (GEM) model. Adjoining histogram shows differences in whole pancreas weights between treatment groups (n=5 mice/arm) at sacrifice; (**F**) Representative images of tumor sections from indicated treatment groups in PKT GEM experiments stained by H&E to demonstrate tumor area (all 20 x, error bar = 20 μm), with comparisons between groups depicted in adjoining histogram. Arrows in the Gem/Pac+αLy6G group show non-malignant epithelial structures. All in vivo experiments were repeated once for reproducibility, and all data points represent biologic replicates. All between-group statistics represent multiple comparison testing using Tukey's post-hoc instrument in one-way ANOVA.

## NLR attenuation during chemotherapy promotes anti-tumor adaptive immunity

In tumor-bearing animals, NLR attenuation significantly reduced—but did not abolish—circulating Ly6G$^+$Ly6C$^{dim}$F4/80$^-$ neutrophilic cells, although a compensatory increase in Ly6C$^{hi}$Ly6G$^-$F4/80$^-$ monocytic cells was observed via flow cytometry from splenocyte-derived CD11b$^+$ cells (*Figure 3A*). As expected, NLR attenuation—either alone or in combination with gemcitabine/paclitaxel—significantly decreased tumor-infiltrating Ly6G/Gr1$^+$ neutrophilic myeloid derived suppressor cells in the PDAC TME (*Figure 3B*). Flow cytometric analysis revealed that treatment with gemcitabine/paclitaxel +anti-Ly6G significantly increased infiltration of both intratumoral CD4$^+$ and CD8$^+$ T-cells, as well as augmented antigen experience (PD-1$^+$) and degranulating capacity (CD107a$^+$) specifically in the CD8$^+$ T-cell compartment (*Figure 3C*), compared with gemcitabine/paclitaxel alone, anti-Ly6G alone, or vehicle arms. Interestingly, increased infiltration in both CD4$^+$ and CD8$^+$ *central* memory (CD44$^+$CD62L$^+$CD103$^-$), but not effector (CD44$^+$CD62L$^-$) or tissue-resident memory (CD44$^+$CD62L$^-$CD103$^+$), T-cells were observed in gemcitabine/paclitaxel +anti-Ly6G-treated tumors compared with gemcitabine/paclitaxel alone, anti-Ly6G alone, or vehicle-treated tumors (*Figure 3D*).

## NLR attenuation during chemotherapy reprograms inflammatory CAF polarization in the PDAC TME

Prior work from our group and others have revealed that inflammatory CAF (iCAF)-derived IL-6 engages in tumor-permissive crosstalk by activating STAT3 signaling with tumor cells (*Nagathihalli et al., 2016*), and that the CAF-tumor cell IL-6/STAT-3 signaling axis is a central mediator of chemoresistance in PDAC (*Hosein et al., 2020*; *Dosch et al., 2021*). Therefore, we investigated if NLR attenuation during chemotherapy improves chemosensitivity by reprogramming iCAF skewness and dampening IL-6/STAT-3 signaling in the TME. Compared with vehicle treatment, concurrent treatment with gemcitabine/paclitaxel +anti-Ly6G (*P*=0.006)—but not anti-Ly6G alone (p=0.12) or gemcitabine/paclitaxel alone (p=0.49) treatment—significantly reduced iCAF (CD45$^-$CD31$^-$PDPN$^+$Ly6C$^+$ MHCII$^-$):myofibroblastic CAF (myCAF; CD45$^-$CD31$^-$PDPN$^+$Ly6C$^-$MHCII$^-$) cellular ratios in vivo (*Figure 4A*; *Appendix 1— figure 5A*). Furthermore, leveraging the near-exclusive expression of *PDPN/Pdpn* in human and murine PDAC-associated CAFs via scRNAseq (*Datta et al., 2022*; *Steele et al., 2020*) (5B&C) and widespread use of PDPN as a pan-CAF marker in multiple PDAC-related studies (*Dominguez et al., 2020*; *Steele et al., 2021*; *Elyada et al., 2019*; *Biffi et al., 2019*; *Neuzillet et al., 2019*), we observed significant reduction in co-expressing PDPN$^+$CXCL1$^+$ stromal cells—presumed iCAFs—in tumors from PKT genetically engineered mice treated with gemcitabine/paclitaxel +anti-Ly6G compared with gemcitabine/paclitaxel alone (p=0.02; *Figure 4B*), validating findings from the KPC orthotopic model.

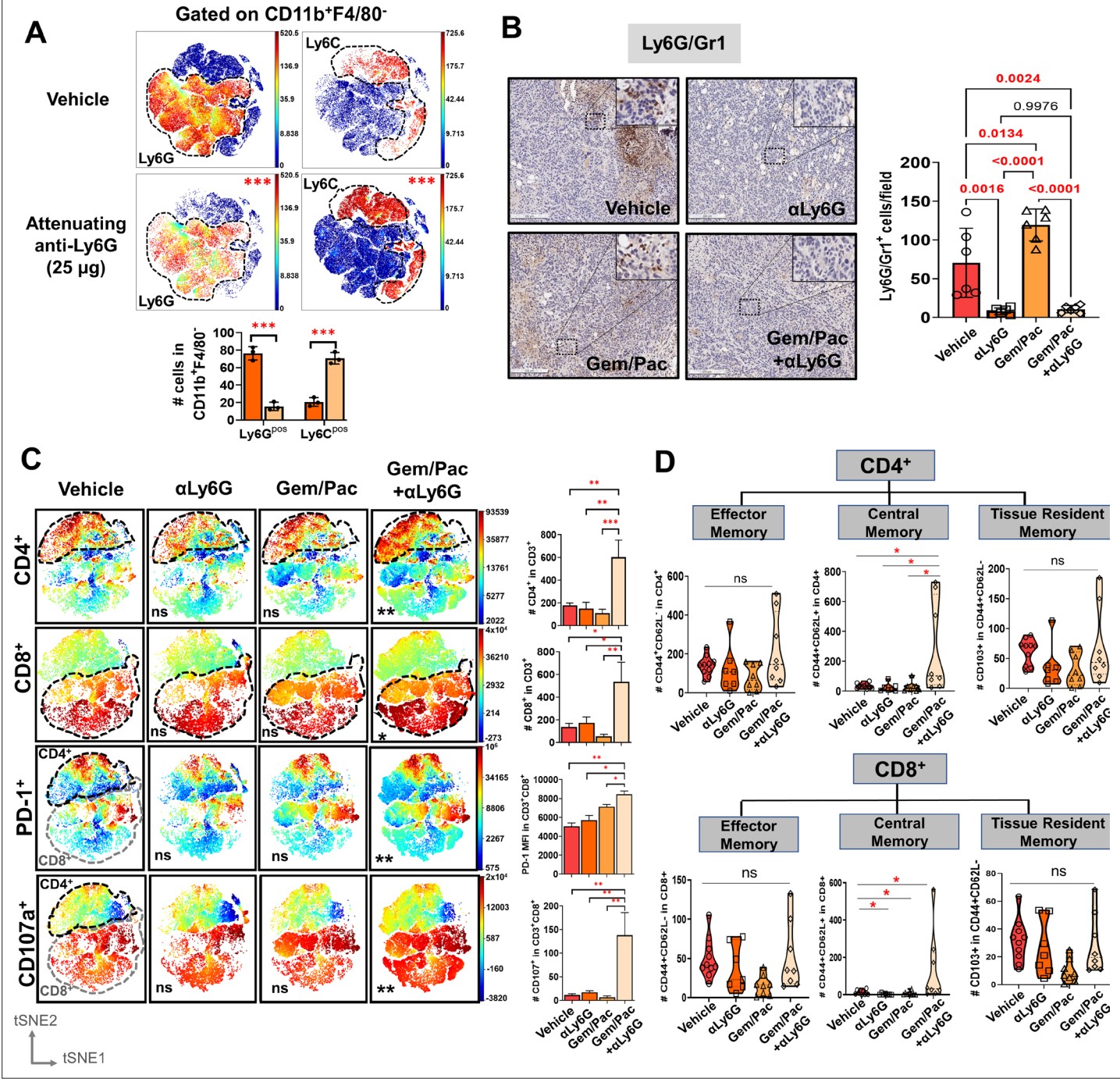

**Figure 3.** Improved chemosensitivity following neutrophil attenuation in pancreatic cancer is associated with anti-tumor adaptive immunity. (**A**) viSNE maps depicting comparison of splenocyte-derived circulating Ly6G+ (*left*) and Ly6C+ (*right*) MDSCs gated within Cd11b+F4/80 cells in NLR-attenuating anti-Ly6G (25 µg) vs. vehicle-treated KPC orthotopic tumor-bearing mice (n=3 mice/arm). Asterisks representing p-values denoting comparisons between anti-Ly6G and vehicle treatment are indicated in the top right left of graph, and quantified in the adjacent histogram; (**B**) Representative images from orthotopic KPC tumor sections in each treatment group stained for Ly6G/Gr1 (all 20 x; scale bar = 200 µm). Ly6G/Gr1+ cells per field are quantified and depicted in the adjacent histogram. High-magnification insets (40 x) indicate relevant areas on these representative sections; (**C**) viSNE maps of total intratumoral CD4+ and CD8+ T-cells gated within CD45+/CD11b-/CD3+ T cells across treatment groups (*top*), and viSNE maps of PD-1+ and CD107a+ in total CD45+/CD11b-/CD3+ T-cells, stratified by CD4+ (*black dotted outline*) and CD8+ (*grey dotted outline*) T-cells by flow cytometry across treatment groups (*bottom;* n=8–10 mice/group). Asterisks representing p-values denoting comparisons between each treatment group and vehicle treatment are indicated in the bottom left of graph. Post-hoc Tukey analysis from one-way ANOVA comparisons between treatment groups for each cell subset are shown in adjoining histograms; (**D**) Violin plots depicting the number of intratumoral effector memory (CD44+CD62L-), central

*Figure 3 continued on next page*

*Figure 3 continued*
memory (CD44+CD62L+CD103-), and tissue-resident memory (CD44+CD62L-CD103+) cells in CD4+ (*left*) and CD8+ (*right*) T-cell compartments across the four treatment arms (n=8–10 mice/group). All experiments were repeated once for reproducibility, and all data points represent biologic replicates; *, p<0.05; **, p<0.01; ***, p<0.001.

## Reduced tissue-level NLR correlates with chemotherapy response, CAF density, and stromal inflammation at single-cell resolution in human PDAC

To examine the association between tissue-level NLR, stromal density/inflammation, and chemotherapy response (partial/complete [n=3], poor/absent [n=3]) in *human* PDAC tumors (*Appendix 1—table 2*) at single-cell resolution, pathologist-selected regions of interest (ROI) from each tumor section probed with metal ion-conjugated antibodies for pancytokeratin (PanCK:epithelial), α-smooth muscle actin (α-SMA:fibroblast), CD11b and CD15 (neutrophil), and CD3 and CD8 (T-cell) were laser-ablated, and atomized ions were acquired using time-of-flight mass cytometry (cyTOF) (*Figure 4C*). Image segmentation and quantification revealed significantly higher ratio of CD11b+CD15+ to CD3+CD8+ cells (NLR; normalized to 5000 total single cells) in pre-treatment tumors from PDAC patients who demonstrated poor/absent pathologic response compared with partial/complete response (15.8±2.8 vs 7.4±3.9; p=0.039) following neoadjuvant chemotherapy (*Figure 4D*). Interestingly, increased NLR in patients with poor/absent pathologic response correlated with significantly higher mean intensity of α-SMA expression (41.9±26.6 vs 18.4±16.6 pixels/cell; p<0.001) in—but not absolute density of—cancer associated fibroblasts in tumor ROIs (*Figure 4D*), as well as relative abundance of co-expressed PDPN+CXCL1+ iCAF populations in corresponding tumor sections (29.7 ± 8.8% vs 18.4 ± 7.4% tumor area; p<0.001; *Figure 4E*).

## NLR attenuation during chemotherapy dampens L-6/STAT-3 signaling in the PDAC TME

The reprogramming of iCAF polarization following NLR attenuation during chemotherapy in the preclinical models was reflected in decreased intratumoral IL-6 and CXCL-1 (data not shown) levels following treatment with gemcitabine/paclitaxel +anti-Ly6G (p=0.0002), but not anti-Ly6G alone (p=0.06) or gemcitabine/paclitaxel alone (p=0.08) treatment, compared to vehicle treatment (*Figure 4F*). In parallel with these findings, compared with vehicle, anti-Ly6G alone, or gemcitabine/paclitaxel alone, gemcitabine/paclitaxel +anti-Ly6G treatment resulted in significantly lower pSTAT3 expression in the tumor cell/epithelial compartment in vivo (p<0.01; *Figure 4G*).

## Neutrophil-derived IL-1β induces pancreatic CAF-tumor cell IL-6/STAT-3 signaling

To explore a mechanistic link between tumor-infiltrating neutrophils and iCAF-mediated IL-6/STAT-3 signaling in the PDAC TME, we characterized the secretome of tumor-infiltrating Ly6G+F4/80- neutrophils isolated from orthotopic KPC tumors, revealing IL-1β as the most robustly secreted cytokine (*Figure 5A*). Systemic NLR attenuation with anti-Ly6G treatment—with or without chemotherapy—resulted in significant diminution of IL-1β secretion in tumor lysates compared with vehicle or chemotherapy treatment in vivo (ANOVA p<0.001; *Figure 5B*), likely due to its incident reduction in systemic and tumor-infiltrating Ly6G+ cells (see *Figure 3*).

Next, we ascertained if neutrophil-derived IL-1β was contributory to CAF-tumor cell IL-6/STAT-3 signaling. Ex vivo co-cultures of KPC CAFs with tumor-infiltrating Ly6G+F4/80- neutrophils derived from orthotopic tumor-bearing KPC mice induced a nearly 20-fold increase in CAF-intrinsic *Il6* transcription (p<0.0001), which was significantly abrogated by either neutralization of IL-1β (p<0.0001) or by pre-incubation of CAFs with IL-1R1 inhibitor Anakinra (*Nagathihalli et al., 2016*; p<0.0001; *Figure 5C*). These results were validated by IL-6 ELISA, which demonstrated a dramatic increase in IL-6 secretion from CAF-neutrophil co-cultures, and was significantly rescued with either IL-1β or IL-1R1 inhibition (all p<0.0001; *Figure 5D*). *Cxcl1* transcription in CAFs—another key iCAF marker—was similarly induced nearly 22-fold following co-culture with tumor-infiltrating neutrophils, and significantly abrogated with either IL-1β or IL-1R1 inhibition (all p<0.0001; *Appendix 1—figure 6*).

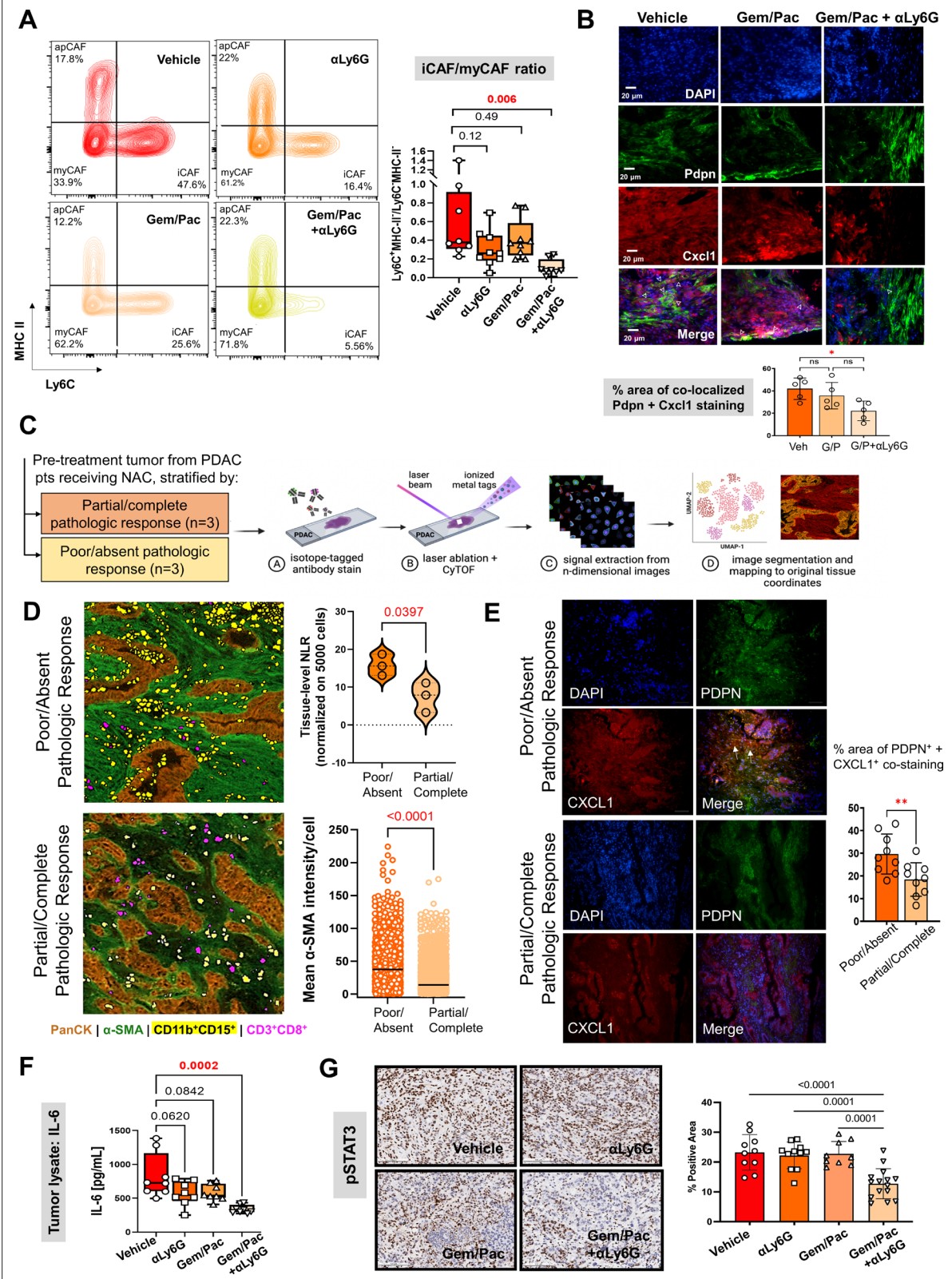

**Figure 4.** Improved chemosensitivity with attenuated NLR is associated with reduction in inflammatory CAF abundance and IL-6/STAT-3 signaling in the tumor microenvironment. (**A**) Representative contour plots of CD45⁻CD31⁻PDPN⁺ cancer-associated fibroblasts (CAF) gated on Ly6C and MHC-II across vehicle, NLR-attenuating αLy6G, gemcitabine plus paclitaxel (Gem/Pac) alone, and Gem/Pac+ αLy6G treatment groups in orthotopic KPC tumor-bearing mice (n=8–10 mice/group) based on percentages of parental cell populations. Inflammatory (iCAF: Ly6C⁺MHC-II⁻), myofibroblastic

*Figure 4 continued on next page*

*Figure 4 continued*

(myCAF: Ly6C⁻MHC-II⁻) and antigen-presenting (apCAF: Ly6C⁻MHC-II⁺) sub-populations are indicated in their respective quadrants. Relative ratios of iCAF/myCAF subsets are quantified in adjacent box-and-whisker plots across treatment groups; (**B**) Immunofluorescent staining for Pdpn (marking CAF), Cxcl1, and merged images (all 20 x; scale bar = 20 μm) from representative tumor sections in PKT mice treated with vehicle, Gem/Pac and Gem/Pac + αLy6G (n=5 mice/arm). Arrows indicate regions with co-localized stromal Pdpn and Cxcl1 staining, with adjacent histogram quantifying % area per section from each biologic replicate with co-localized stromal Pdpn and Cxcl1 staining; (**C**) Schematic representation of imaging mass cytometry (IMC) workflow to provide spatially resolved single-cell phenotypes of human PDAC tumors derived from pre-treatment specimens which underwent neoadjuvant chemotherapy and ultimately demonstrated partial/complete or poor/absent pathologic response (n=3 each); (**D**) Single-cell segmentation of CD11b⁺CD15⁺ neutrophils and CD3⁺CD8⁺ T-cells mapped onto representative tissue section from tumors showing poor/absent and partial/complete response, with epithelial (PanCK) and stromal (α-SMA) territories also shown. Adjacent violin plot (*top*) quantifies tissue-level neutrophil-to-lymphocyte (NLR) across three tumors in each group, calculated as #CD11b⁺CD15⁺ ÷ #CD3⁺CD8⁺ cells (normalized to 5000 total single cells), while histogram (*bottom*) tabulates mean pixel intensity of α-SMA expression in stromal cells across three tumors each in partial/complete vs. poor/absent responder cohorts; (**E**) Immunofluorescent staining for PDPN (marking CAF), CXCL1, and merged images (all 20 x) in representative sections from the same tumors shown in (**D**) stratified by poor/absent vs. partial/complete response. White arrows indicate regions with co-localized stromal PDPN and CXCL1 staining, with adjacent histogram quantifying % area with co-localized stromal PDPN and CXCL1 staining. For the latter comparison, three separate sections from each biologic replicate (n=9 total sections) were used for latter comparison; (**F**) Quantification of IL-6 ELISA (pg/ml) from whole tumor protein lysates across vehicle, αLy6G-treated, gemcitabine +paclitaxel (Gem/Pac) alone-treated, and Gem/Pac+αLy6G-treated orthotopic KPC tumor-bearing mice (n=8–10 mice/group); (**G**) Representative images from tumor sections in each treatment group (n=5 mice/group) stained for phospoSTAT3 (all 20 x; scale bar = 200 μm), and adjacent bar graph showing quantification of % positive area of pSTAT3 in the epithelial compartment per field. All between-group statistics represent multiple comparison testing using Tukey's post-hoc instrument in one-way ANOVA. When absolute p-values not provided: *, p<0.05; **, p<0.01.

Finally, KPC tumor cells demonstrated significantly higher pSTAT3 expression when incubated with conditioned media (CM) from intratumoral neutrophil-CAF co-cultures alone compared with CM from neutrophil-CAF co-cultures treated with either anti-IL-1β or anti-IL-6 neutralizing antibodies (*Figure 5E*). Together, these data reveal a role for neutrophil-derived IL-1β in promoting iCAF polarization and inducing CAF-tumor cell IL-6/STAT3 signaling in the PDAC TME, which is a central mediator of chemoresistance (*Figure 5F*).

## Discussion

In selected patients with operable pancreatic cancer undergoing curative-intent pancreatectomy following modern chemotherapy, we identify for the first time that NLR dynamics *during* NAC correlate strongly with pathologic response, and an NLR score encompassing these dynamics is prognostic of disease-free and overall survival. While these novel findings warrant large-scale multi-institutional validation to strengthen and/or reconcile data from heterogeneous PDAC populations (*Hasegawa et al., 2016*; *Kubo et al., 2019*; *Strong et al., 2022*), the present data indicate that both baseline NLR *and* NLR dynamics may be promising metrics of response and overall disease trajectory in patients with localized PDAC, recapitulating evidence from other gastrointestinal cancers (*Sato et al., 2012*).

The relationship between systemic chemotherapy, ensuing cytotoxicity/tumor-cell death and its immune repercussions, neutrophil mobilization and trafficking, adaptive immune dysfunction, and clinical outcomes in solid tumors is complex (*Banerjee et al., 2013*). Notwithstanding, since systemic chemotherapy does not appear to impact tumor-infiltrating neutrophils (*Nywening et al., 2018*) or circulating NLR in our preclinical studies, these data also suggest that therapeutic strategies to attenuate NLR *during* NAC may improve pathologic response in operable PDAC.

While the etiologies underlying the attenuation of *endogenous* NLR in patients demonstrating decreasing ΔNLR during NAC in this study are undoubtedly complex and remain unclear, modeling this phenomenon in preclinical models suggests that a 'priming' phase in which the systemic NLR is *actively* dampened improves chemosensitivity and is associated with heightened adaptive anti-tumor immunity in the PDAC TME. In our preclinical modeling, attenuation of NLR immediately preceding and during gemcitabine/paclitaxel chemotherapy not only improved CD4⁺ T-helper and CD8⁺ T-effector cell trafficking, but also amplified CD4⁺/CD8⁺ central memory skewness as well as CD8⁺ T-cell antigen experience and degranulating capacity. Our data add nuance to previous findings indicating that depletion of Ly6G⁺ neutrophilic myeloid-derived suppressor cells unmasks adaptive immunity (*Stromnes et al., 2014*), or that ablation of CXCR2⁺ tumor-associated neutrophils augments IFN-γ⁺CD8⁺ T-cell infiltration to potentiate FOLFIRINOX responses in PDAC models (*Nywening*

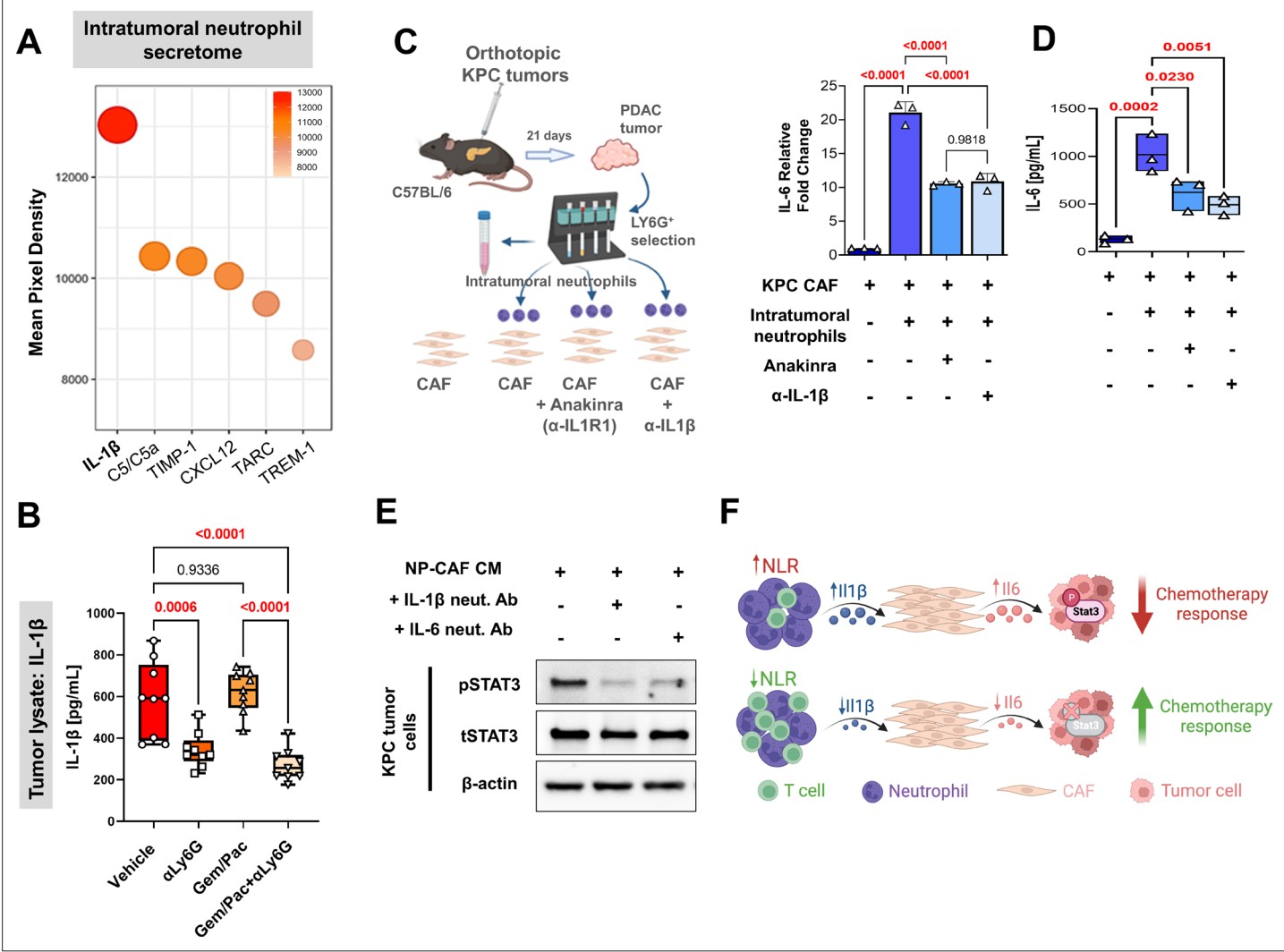

**Figure 5.** Neutrophil-derived IL-1β induces pancreatic fibroblast-tumor cell IL-6/STAT-3 signaling. (**A**) Bubble plot representing multiplex cytokine array performed on condition media from column-sorted Ly6G+F4/80- neutrophils (24-hr culture) derived from whole pancreata of KPC orthotopic mice. The chemiluminescent intensity of the six most robustly expressed cytokines is quantified as mean pixel density; (**B**) Quantification of IL-1β ELISA (pg/ml) from whole tumor protein lysates from vehicle, NLR-attenuating αLy6G, gemcitabine plus paclitaxel (Gem/Pac) alone, and Gem/Pac+ αLy6G treatment groups in orthotopic KPC tumor-bearing mice (n=9 mice/group); (**C–D**) Schematic of experimental design illustrating ex vivo co-culture of KPC CAFs with intratumoral column-sorted Ly6G+F4/80 cells from whole pancreata of KPC orthotopic mice, with or without pre-treatment of CAFs with anakinra (α-IL1R1 antibody) or pre-treatment of neutrophils with α-IL-1β neutralizing antibody (*left*); (**C**) qPCR analysis representing relative fold change in *Il6* gene expression, and (**D**) quantification of IL-6 ELISA (pg/ml) from conditioned media collected from co-culture conditions comparing CAFs alone with CAFs co-cultured with intra-tumoral neutrophils with or without anakinra or α-IL-1β antibody pre-treatment. Results show mean ± SEM of three biologic replicates; (**E**) Western blot analysis of pSTAT3^Y705 and total STAT3 (tSTAT3) levels from KPC tumor cell lysates following incubation with conditioned media (CM) from ex vivo intratumoral neutrophil (NP)-CAF co-cultures, either alone or treated with anti-IL-1β or IL-6 neutralizing antibodies. All experiments were repeated once for reproducibility, and all data points represent biologic replicates. All between-group statistics represent multiple comparison testing using Tukey's post-hoc instrument in one-way ANOVA; (**F**) Graphical summary of proposed neutrophil-CAF-tumor cell IL-1β/IL-6/STAT-3 signaling axis that underlies the associated between NLR dynamics and chemotherapy response in PDAC. When absolute p-values not provided: *, p≤0.05; **, p≤0.01; ***, p≤0.001.

*et al., 2018*). Given that systemic neutrophilic *silencing* is not only clinically impractical, but also drives a compensatory and dynamic myelopoiesis (e.g. of CCR2+ macrophages) that thwarts anti-tumor immunity (*Nywening et al., 2018*), the chemosensitizing and immune-potentiating effects of NLR *attenuation* in our model may be related to the disruption of specific tolerogenic functions inherent to tumor-associated neutrophils. Indeed, ongoing investigation in our laboratory is focused

on deciphering and targeting neutrophil-intrinsic tolerogenic mechanisms that orchestrate immuno-suppressive tumor-stromal-immune crosstalk and promote therapeutic resistance in PDAC.

One such potential mechanism governing therapeutic resistance unveiled in the present study is the previously unrecognized role of neutrophil-derived IL-1β in driving iCAF polarization and CAF-tumor cell IL-6/STAT-3 signaling in the PDAC TME. As such, the improved chemosensitivity associated with NLR attenuation in our preclinical models suggest that combining chemotherapy with therapeutic strategies to mitigate neutrophil-stromal-tumor cell IL-1β/IL-6/STAT-3 signaling in PDAC patients may be advantageous. Results from the Precision Promise[SM] trial investigating anti-IL-1β antagonism in combination with gemcitabine/*nab*-paclitaxel and PD-1 inhibition in patients with advanced PDAC (NCT04581343) are eagerly awaited. Ultimately, decoding the intersection between NLR dynamics, the balance between tumor-permissive inflammation and anti-tumor adaptive immunity, and tumor-stromal-immune cellular crosstalk that perpetuates chemoresistant signaling circuitries in PDAC may lay the foundation for novel interventions to overcome chemotherapy resistance and improve contemporary outcomes in this lethal malignancy.

## Grant support

Supported by KL2 career development grant by Miami CTSI under NIH Award UL1TR002736, Stanley Glaser Foundation, American College of Surgeons Franklin Martin Career Development Award, and Association for Academic Surgery Joel J. Roslyn Faculty Award (to J. Datta); NIH R01 CA161976 (to N.B. Merchant); and NCI/NIH Award P30CA240139 (to J. Datta and N.B. Merchant).

## Acknowledgements

The authors wish to acknowledge Xizi Dai, Samara Singh, and Austin Dosch for input during the preclinical experiments in this project. Funding: Supported by KL2 career development grant by Miami CTSI under NIH Award UL1TR002736, Stanley Glaser Foundation, American College of Surgeons Franklin Martin Career Development Award, and Association for Academic Surgery Joel J Roslyn Faculty Award (to J Datta); NIH R01 CA161976 (to NB Merchant); and NCI/NIH Award P30CA240139 (to J Datta and NB Merchant).

## Additional information

### Funding

| Funder | Grant reference number | Author |
|---|---|---|
| National Institutes of Health | UL1TR002736 | Jashodeep Datta |
| American College of Surgeons | Franklin H. Martin Research Fellowship | Jashodeep Datta |
| Association for Academic Surgery Foundation | Joel J. Roslyn Faculty Award | Jashodeep Datta |
| University of Miami | Stanley Glaser Foundation Award | Jashodeep Datta |
| National Cancer Institute | P30CA240139 | Nipun B Merchant Jashodeep Datta |
| National Cancer Institute | R01CA161976 | Nipun B Merchant |

The funders had no role in study design, data collection and interpretation, or the decision to submit the work for publication.

### Author contributions

Iago de Castro Silva, Data curation, Formal analysis, Investigation, Visualization, Methodology, Writing - original draft, Writing - review and editing; Anna Bianchi, Data curation, Software, Formal analysis, Investigation, Visualization, Methodology, Writing - original draft, Writing - review and editing; Nilesh U Deshpande, Data curation, Formal analysis, Investigation, Visualization, Methodology, Writing

- review and editing; Prateek Sharma, Conceptualization, Data curation, Investigation, Methodology, Writing - review and editing; Siddharth Mehra, Data curation, Investigation, Visualization, Methodology, Writing - review and editing; Vanessa Tonin Garrido, Data curation, Software, Formal analysis, Investigation, Writing - review and editing; Shannon Jacqueline Saigh, Data curation, Software, Formal analysis, Methodology, Writing - review and editing; Jonathan England, Resources, Data curation, Formal analysis, Investigation, Writing - review and editing; Peter Joel Hosein, Data curation, Validation, Investigation, Writing - review and editing; Deukwoo Kwon, Data curation, Formal analysis, Investigation, Writing - review and editing; Nipun B Merchant, Resources, Data curation, Supervision, Funding acquisition, Validation, Investigation, Visualization, Methodology, Writing - review and editing; Jashodeep Datta, Conceptualization, Resources, Data curation, Software, Formal analysis, Supervision, Funding acquisition, Validation, Investigation, Visualization, Methodology, Writing - original draft, Project administration, Writing - review and editing

### Author ORCIDs
Iago de Castro Silva ⓘ http://orcid.org/0000-0002-3898-6173
Jashodeep Datta ⓘ http://orcid.org/0000-0003-2869-1571

### Ethics
Human subjects: -Informed consent was not necessary since tissue sections were accrued previously under a cancer center-wide biospecimen protocol -Institution Review Board /PRMC/site disease group approval for this study was obtained under protocol 20200123.

This study was strictly performed in agreement with all the recommendations stablished in the Guide for the Care and Use of Laboratory Animals of the National Institutes of Health. All animal work was performed following the approved Institutional Animal Care and Use Committee (IACUC) protocol (#21-057) of the University of Miami, and supervised by the Division of Veterinary Resources (DVR). All surgical procedures were performed under general anesthesia, analgesic drugs were administered postoperatively, and every effort was made to minimize any form of animal suffering.

### Decision letter and Author response
Decision letter https://doi.org/10.7554/eLife.78921.sa1
Author response https://doi.org/10.7554/eLife.78921.sa2

## Additional files

### Supplementary files
- MDAR checklist
- Source data 1. Clinical Cohort and Western Blotting Source Data.

### Data availability
Clinicodemographic data utilized in this analysis can be tracked back to individual patients (e.g., age, CA19-9 values, pathologic response score) despite deidentification. Since these comprise protected health information from human subjects, a limited deidentified dataset with only relevant data to allow reproduction of major findings are provided. All relevant source data from in vitro experiments have also been provided.

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

# Appendix 1

## A. Detailed and supplementary methods

### Clinical cohort

After obtaining Institutional Review Board approval for the study, patients with localized pancreatic ductal adenocarcinoma (PDAC) who received neoadjuvant chemotherapy (NAC) with either mFOLFIRINOX, gemcitabine/abraxane, or both and underwent pancreatectomy between 7/2015 and 12/2019 at a tertiary academic medical center were enrolled. This study duration was chosen due to the near-routine use of modern multi-agent chemotherapy in the neoadjuvant setting for potentially operable PDAC at our institution beginning in 2015. Resectability criteria for patients were adjudicated by individual surgeons in our practice and/or in multidisciplinary discussion using the National Comprehensive Cancer Network resectability framework.

Complete blood count (CBC) data was accrued on 94 patients enrolled in the study at two timepoints: (1) prior to initiation of neoadjuvant chemotherapy (pre-NAC); and (2) prior to surgical resection (pre-surgery). Specific elements of the CBC tabulated were total leukocyte count, proportion of neutrophils, proportion of lymphocytes, and platelet count. Extreme care was taken to ensure that the CBC accrued for analysis was obtained in the absence of cholangitis or other infection, and that no concurrent antibiotics were being administered. Moreover, we ensured that no patient included in the study was actively on immunosuppressive (e.g. steroids) or immunomodulatory (e.g. TNF inhibitors) agents. Utilization of filgastrim or G-CSF (Neupogen) during NAC was recorded—all but three patients (mFOLFIRINOX, n=1; gemcitabine/nab-paclitaxel, n=2) received G-CSF support during NAC to counteract neutropenia. Other clinical variables accrued have been defined previously *Macedo et al., 2019* and are provided in *Appendix 1—table 1*.

Perioperative information and data about the type and timing of NAC, including regimen, number cycles/months, and RT modalities were collected as described previously (*Macedo et al., 2019*). The final analysis only included patients who underwent NAC with mFOLFIRINOX, gemcitabine/abraxane, or both with or without radiotherapy followed by curative-intent pancreatectomy. The choice of chemotherapy regimen and dosage was selected at the discretion of the medical oncologist.

All patients included in this study underwent surgical resection following NAC. Surgically resected specimens were then evaluated by board-certified gastrointestinal pathologists at the University of Miami, and classified into the following grades as defined by the College of American Pathology (*Dosch et al., 2021*) and as described previously (*Macedo et al., 2019*): pCR or *grade 0* (ypT0/N0) was defined as no evidence of primary tumor or any nodal involvement in the final surgical specimen. No patient with pCR was included in this study either due to lack of follow-up and/or incomplete clinical annotation. Partial pathological response (pPR) included both *grade 1* (minimal residual cancer with single cells or small groups of cancer cells) and *grade 2* (residual cancer outgrown by fibrosis). *Grade 3* or poor/absent pathological response was defined when extensive residual cancer was identified in the surgical specimen. The study cohort was then dichotomized into '*complete/ partial*' (grades 0–2) and '*poor/absent*' *response (grade 3)* for analysis.

The primary endpoints were pathologic response, disease-free (DFS), and overall survival (OS). OS was measured from the time from the date of diagnosis until death from any cause. DFS was calculated from the time of surgical resection until the date of recurrence or death to avoid immortal time bias.

### Patient and public involvement

Patients or the public were not involved in the design, or conduct, or reporting, or dissemination plans of our research.

### Statistical analysis of clinical data

Descriptive statistics were calculated for patients' characteristics using median, interquartile range (IQR), frequencies, and 95% confidence intervals (95% CIs). Both pre-NAC absolute NLR as well as fluctuation of NLR during NAC, defined as ΔNLR = pre-surgery NLR—pre-NAC NLR was tabulated. The association between complete/partial and poor/absent pathologic response following NAC and the study variables (clinical variables, aNLR, and ΔNLR) were examined using univariable and multivariable logistic regression models while controlling for demographic (age, gender, BMI), clinical (resectability status, tumor size, any CA19-9 decline [i.e. any decrease vs. stability or any

increase during NAC]), and treatment-related (duration of NAC, radiotherapy receipt) variables. No pathologic or post-surgical variables were included in this model. Adjusted odds ratios (ORs) and corresponding 95% confidence intervals and p-values were reported.

Area under the receiver operating characteristic curves (AUCs) were estimated for three biomarker models (pre-chemotherapy aNLR only, ΔNLR only, combined model aNLR + ΔNLR). Internal validation using bootstrap logistic regression—wherein 1,000 random samples were generated with replacement—was conducted to obtain mean AUCs and corresponding 95% confidence intervals. An interaction test corroborated the incremental improvement of the combined aNLR + ΔNLR model (vs. aNLR or ΔNLR only) in predicting pathologic response (ΔNLR vs. combined ΔNLR +aNLR, <0.001; aNLR vs. combined ΔNLR +aNLR, p=0.049).

From the combined biomarker model, we calculated an 'NLR score' comprising the product of regression coefficients and aNLR/ΔNLR = *10.45–2.9224\*aNLR-2.13\*ΔNLR*. Estimated scores were examined for three biomarker models of disease-free survival (DFS) and overall survival (OS) with respect to time-dependent AUC between 1 year and 3 years. We also conducted internal validation for time-dependent AUC and reported mean AUC at 1 year and 2 years along with 95% confidence intervals in time-specific receiver operating characteristic curve (data not shown). We dichotomized the 'NLR score' at zero (score >0 vs score ≤0) and compared DFS and OS using log-rank test in Kaplan-Meier curves. Hazard ratio (HR) was estimated using Cox proportional hazards regression model. All tests were two-sided and statistical significance was considered when ≤0.05.

## Imaging mass cytometry of human PDAC tumors
### Staining

Human PDAC FFPE sections were mounted on glass slides, heated in dry oven at 62 °C for 2 hr, and deparaffinized in xylene followed by rehydration using descending grades of ethanol. Antigen retrieval was performed using Tris-EDTA buffer as previously described[1], prior to incubation with BlockAidTM Blocking Solution (ThermoFisher Scientific, #B10710). Tissue sections were stained overnight with the antibody cocktail (all Fluidigm: **Cell-ID**, 191-193Ir, Cross-species, 201192 A; **CD3**, 170Er, Human, 3170019D; **CD11B**, 149Sm, Human, 3149028D; **α-SMA**, 141Pr, Human, 314017D; **Pan-Cytokeratin**, 148Nd, Human, 3148022D; **CD15**, 164Dy, Human, 3164001B; **CD8**, 146Nd, Human, 3146001B) at 4 °C in a humid chamber. Slides were then washed in 0.2% Triton X-100 and incubated with DNA intercalator for 30 minutes at RT. Slides were then washed and air-dried.

### VisioPharm analysis

Samples were acquired using the Hyperion Imaging Mass Cytometer (Standard Biotools Inc) and the resulting.mcd data files were transformed in MCD Viewer (v 1.0.560.6, Standard Biotools Inc), exported as a 32-bit multi-page OME-TIFF file without rescaling, and analyzed with the Visiopharm Image Analysis module (v 2022.07.0.12224, Visiopharm Corporation). Regions of Interest (ROI) were drawn on the image, and cell identification and segmentation were performed using the pretrained 'Nuclei Detection, AI (Fluorescence)' APP provided with the software. The APP was modified to allow detection of nuclei in IMC data by changing the input band from DAPI to 193Ir. After cells in the image were identified, a new APP was created to determine the cell phenotypes present within the ROI, using the embedded 'Phenotype' Classification Method. Both the background and upper levels of staining were determined 'Features' function for each marker, and then used in the 'Setup' section of the Phenotype classification to establish the range of positive pixel intensities for each marker. The APP was then trained to produce multiple phenotype classes. The resulting phenotypes were manually verified for each marker and, if inaccurate, the 'Setup' values were modified, and the APP was retrained. This process was repeated until the phenotypes identified matched the manually identified marker for each layer. Once verified T-cells and neutrophils phenotypes were combined into one main classification using post processing steps within the APP. In the final step of the APP creation, data output variables were created. This included the mean intensity, median, and standard deviation of the pixel intensity within each defined 'object' (cell). In addition, Center X and Center Y of the multiplexing values were added to the data output. The APP was then run, and a visual spatial phenotyping map was created. All the data outputs were also exported from the program as.tsv files which were converted in Excel to.xlsx files for further downstream analysis.

## Orthotopic murine PDAC model generation

All animal experiments were performed in accordance with the NIH animal use guideline and protocol (#21–057) approved by the Institutional Animal Care and Use Committee (IACUC) at the University of Miami. C57BL/6 female mice (6 weeks age) were purchased from Jackson Laboratory (Bar Harbor, ME) and housed at the institutional animal facility under constant temperature and humidity, on a 12 hr light and dark cycle, with available standard food and filtered water. For orthotopic model generation, C57BL/6 mice were anesthetized with Ketamine:Xylazine (10:1) in sterile 0.9% NaCl saline. The abdomen was then wiped with iodine-based solution followed by alcohol wipes. The mice were placed on a clean sterile pad onto a warming pad. A paramedian incision was made in the abdomen using scissors and the pancreas and spleen were externalized with assistance of curved forceps. Ten μL of cell suspension containing syngeneic $50 \times 10^3$ KPC6694c2 (henceforth referred to as KPC) tumor cells in Matrigel were injected directly into the pancreas using a Hamilton microliter syringe. The pancreas and spleen were then returned to the intra-abdominal cavity in the anatomical position and the peritoneum sutured with 5–0 Vicryl suture, and the skin closed with skin staples. Immediately after surgery, 300 μL of sterile saline was injected subcutaneous for fluid maintenance and buprenorphine 0.05–0.1 mg/kg was administered subcutaneously for pain management. Mice were continuously monitored post-operatively and staples were removed one week after surgery.

## In vivo experimental design

For the initial anti-Ly6G titration experiment, C57BL/6 female mice (n=3 each) were orthotopically injected with KPC tumor cells and 10 days following surgery, treatment was initiated with intraperitoneal Ly6G neutralizing antibody (BioXCell clone 1A8) q3 days for 2 weeks (n=3/arm): vehicle, 25 μg dose, 100 μg dose and 200 μg dose; the latter is a well-established depleting dose. Our specific intention was to attenuate, but not deplete, systemic neutrophils to simulate the endogenous decline in ΔNLR during neoadjuvant chemotherapy. After sacrifice, blood and spleens were collected for flow cytometry analysis.

The anti-Ly6G clone 1A8 neutralizing antibody construct is a rat IgG2a that induces a Fc-dependent opsonization and phagocytosis of Ly6G$^+$ cells. To demonstrate the specificity of this antibody, a separate experiment showed that NLR-attenuating anti-Ly6G treatment in KPC orthotopic tumor-bearing mice specifically reduced splenic (data not shown) and intratumoral CD11b$^+$F4/80$^-$Ly6G$^+$ neutrophils, but not other granulocytic populations—namely CD11b$^+$F4/80$^-$Siglec-F$^+$ eosinophils and F4/80$^-$CD11c$^-$FcεR1$^+$ basophils—compared with vehicle treatment (*Appendix 1—figure 7*).

In the main in vivo experiments, C57BL/6 mice were orthotopically injected with KPC tumor cells and randomized into four treatment groups after 10 days of tumor growth (n=8–10/arm): vehicle control, anti-Ly6G antibody treatment alone with neutrophil-attenuating—but not depleting—dosing (25 μg/dose; see *Appendix 1—figure 3A*) every 3 days, gemcitabine (100 mg/kg) and paclitaxel (10 mg/kg) once weekly (Targetmol), and gemcitabine/paclitaxel plus anti-Ly6G. Mice were sacrificed following 3 weeks of treatment, tumor burden evaluated, and tumors subjected to histological analysis and immunophenotyping by flow cytometry. Tumor-bearing *Ptf1a*$^{Cre/+}$*Kras*$^{LSL-G12D/+}$*Tgfbr2*$^{flox/flox}$ (PKT) mice were generated as previously described. PKT mice were randomly assigned to three treatment groups at 4 weeks of age (n=5/group): vehicle control, gemcitabine (100 mg/kg) plus paclitaxel (10 mg/kg) chemotherapy alone both once weekly, and neutrophil-attenuating anti-Ly6G antibody treatment q3 days plus gemcitabine/paclitaxel once weekly. Mice were sacrificed after 2 weeks of treatment for endpoint analysis of tumor weights and histopathologic readouts. In vivo experiments were repeated to ensure reproducibility.

I.D.S, A.B., N.U.D, and J.D. were aware of the randomization schemes as well as blinding procedures for mouse allocation and experimental conduct. There were no exclusions. All reporting of in vivo experiments followed the ARRIVE guidelines.

## Flow cytometry

Flow cytometric analysis was performed as described previously (*Nagathihalli et al., 2016*; *Dosch et al., 2021*; *Datta et al., 2022*). Briefly, whole pancreata harvested from C57BL/6 mice were digested enzymatically utilizing a solution containing 0.6 mg/ml of collagenase P (Roche), 0.8 mg/ml Collagenase V (Sigma Aldrich), 0.6 mg/ml soybean trypsin inhibitor (Sigma Aldrich), and 1800 U/ml DNase I (ThermoFisher Scientific) in RPMI medium for 20–30 min at 37 °C. Samples were then washed and underwent single cell dissociation using 40 μm smash strainers. Spleens obtained from the same animals were passed through 100 μm mesh filters and both blood and spleen single cell

suspensions were processed using RBC lysis buffer. Samples were then kept frozen at –80 °C. Prior to flow cytometry staining, samples were thawed, washed and incubated with FcR blocking reagent (Miltenyi Biotec), and subsequently stained with fluorescently conjugated antibodies as follows: CD45 (FITC, Biolegend, Catalog #103108 or BV510, Biolegend 103138 or BUV805, BD 741957); CD3 (PerCPCy5.5, Biolegend 100218); CD4 (BV785, Biolegend 100453 or Pac Blue Biolegend 100428); CD8 (BV605, Biolegend 100744 or BV510, Biolegend 100752); CD44 (APC-Cy7, Biolegend 103028); CD62L (PE, Biolegend 104408); PD-1 (PE-Cy7, Biolegend 135216); CD107a (PE-Dazzle, Biolegend 121624); CD11b (BUV805, BD 741934 or BV750, Biolegend 101267); PDPN (BV421, Biolegend 127423); MHC-II (BV711, Biolegend 107643 or APC-Fire750, Biolegend 107652); EpCAM (PE 118206, Biolegend), Ly6C (PE-Cy7, Biolegend 128018), CD31 (APC-Cy7, Biolegend 102534), Ly6G (PerCP, Biolegend 127654 or BUV563, BD 612921); FceR1 (PE, ThermoFisher 12-5898-82); CD19 (PE-Cy5.5, ThermoFisher 35-0193-82); Nk1.1 (PE-Dzz594, Biolegend 108748); Siglec F (APC, BD 562680), F4/80 (AF488, Biolegend 123120); CD11c (AF700, Biolegend 117320); and TCRb (BUV661, BD 749914). Ghost Red Dye 780 (TONBObiosciences) live/dead cell discrimination was performed as per manufacturer's protocol and cells were fixed with 1% formaldehyde solution (ThermoFisher).

Flow cytometry data acquisition was performed on CytoFLEX S (Beckman Coulter) and analyzed using FlowJo v10 software (BD Life Sciences). For flow cytometry data visualization, viSNE maps were created using Cytobank software (Beckman Coulter) and all biological replicates (n=8–10 mice/ group) were concatenated and merged into a single sample.

## Single-cell RNA sequencing in PKT mice

Single-cell RNA sequencing in vehicle-treated PKT mice has been reported previously (*Datta et al., 2022*). The 10 x Genomics Chromium Single Cell 3' Reagent v3.1(Cat # PN-1000268) was used with standard conditions and volumes to process cell suspensions for 3' transcriptional profiling. Single-cell suspensions from PKT mice were extracted, and live cells were sorted using flow cytometry, and volumes were calculated for a target cell recovery of 100,000 cells and loaded on the Chromium Controller as per manufacturer guidelines. The resultant purified cDNA was quantified and qualitative assessed on the Agilent Bioanalyzer using the High Sensitivity DNA Kit (Cat #5067–4626). The final single-cell 3' libraries were quantified using the Qubit dsDNA High Sensitivity (Cat #Q33231) and qualitatively evaluated on the Agilent Bioanalyzer using the High Sensitivity DNA Kit. For sequencing, libraries were loaded at optimized concentrations onto an Illumina NovaSeq and paired end sequenced under recommended settings (R1: 28 cycles; i7 index: 10 cycles; i5 index: 10 cycles; R2: 90 cycles). The libraries were diluted to varying nM concentrations in Illumina Resuspension Buffer (PN-15026770), denatured according to Illumina standard guidelines, and loaded on the Illumina NovaSeq at 1.2 nanomolar. The resulting intensity files were demultiplexed as FASTQ files using Illumina BaseSpace software and then aligned to the transcriptome using the 10 x Genomics CellRanger (ver4.0.0) software package.

Principal component analysis was performed on the scaled data to reduce the dimensions, with number of components chosen based on a cumulative proportion (accumulated amount of explained variance) of 90%. Cells with >5% mitochondrial counts, less than 200 or more than 2500 unique feature counts were filtered out. Clusters of cells were identified by shared nearest neighbor algorithm. Cell type annotations were assigned to clusters based on the expression of canonical features in a minimum percentage of cells. For this analysis, CAFs were nominated by exclusive expression of *Col1a1* and *Col1a2* (see *Appendix 1—figure 5*).

## Ex vivo cancer-associated fibroblast (CAF)-neutrophil co-culture experiments

Cell suspension of whole pancreas obtained from KPC orthotopic mice (same procedure as described above) underwent magnetic column separation to obtain Ly6G+ cells as per mouse Myeloid-Derived Suppressor Cell Isolation Kit manufacture's protocol (Miltenyi Biotec). Ly6G+ cells were then co-cultured with KPC CAFs (*Nagathihalli et al., 2016*; *Park et al., 2020*) in standard six-well plates. The conditions of the experiment were: (1) CAFs alone ($2\times10^5$ cells per well); (2) CAFs +intra-tumoral Ly6G+ cells ($10^6$ cells per well)' (3) CAFs pre-treated with Anakinra (α-IL-1R1 inhibitor; with license from SOBI Pharmaceuticals, Sweden) 50 µg/mL +intra-tumoral Ly6G+ cells; and (4) CAFs +intra-tumoral Ly6G+ cells treated with anti-IL-1β neutralizing antibody 10 µg/mL (R&D Systems cat# AF-401-NA). Each condition was performed in three biological replicates. After 24 hr of co-culture, condition media was collected, and RNA obtained from CAFs using the RNeasy Kit (Qiagen) according to the manufacturer's protocol. RNA concentration and quality were verified using NanoDrop spectrophotometer (ThermoFisher). cDNA was then generated by reverse transcription of RNA product using High-Capacity cDNA Reverse Transcription Kit with RNase

Inhibitor (Applied Biosystems). Quantitative polymerase chain reaction (qPCR) was performed using incubation with *Il6* and *Cxcl1* mouse gene-specific predesigned primers (RT2 qPCR Primer Assay, Qiagen) and iQ SYBR Green Supermix (Bio-Rad). Gene expression was normalized to the housekeeping gene GAPDH using the ΔΔCT method and reported as fold change relative to control.

KPC6694c2 tumor cells were incubated with conditioned media harvested from ex vivo co-cultures of intratumoral neutrophils and CAFs as described above, either alone or with anti-IL-1β or anti-IL-6 neutralizing antibodies (Thermofisher, Waltham, MA), and ensuing protein lysates blotted for phosphorylated STAT-3 (pSTAT3) and total STAT-3, as described previously (*Nagathihalli et al., 2016*; *Dosch et al., 2021*; *Datta et al., 2022*). Total STAT3 (124H6) and pSTAT3 (Y705) antibodies were purchased from Cell Signaling Technology (Denver, MA).

## Enzyme-linked immunosorbent assay (ELISA) and multiplex cytokine array

Condition media obtained from the ex vivo co-culture conditions (see above) was harvested, and cellular contaminants were removed by centrifugation at 4 °C. For whole pancreatic tumor lysates, 5–10 mg of tumor or normal pancreatic tissue was lysed in 1 x RIPA buffer, and cellular contaminants were removed by centrifugation at 4 °C. Lysates were quantified and equal amount of protein loaded per well, including three technical replicates per sample. ELISA kits specific for mouse IL-6, CXCL-1, and IL-1β (R&D Systems) were used according to the manufacturer's protocol to quantify levels of those cytokines in the condition media and intra-tumoral protein lysates.

Condition media from column-sorted intra-tumoral Ly6G+ cells cultured for 24 hr was utilized to perform a Proteome Profiler Mouse Cytokine Array Kit per manufacturer protocol (R&D Systems). Results were analyzed by measuring pixel intensity of corresponding cytokine dots using HLImage ++software (Western Vision).

## Histologic Analysis

Pancreatic tissues were fixed in 10% neutral-buffered formalin followed by 70% ethanol after 24 hr and embedded in paraffin. For hematoxylin and eosin (H&E) staining, sections were mounted on glass slides and deparaffinized in xylene followed by rehydration using alcohol gradient. Thereafter, slides were stained with hematoxylin solution for 3 min and eosin solution for 1 min and washed in between with running tap water. H&E sections from each group were then evaluated by a board-certified GI pathologist at the University of Miami (J.E.). The pathologist remained blinded to the treatment groups.

Immunohistochemistry (IHC) staining was performed by HistoWiz Inc Briefly, sections were also mounted on glass slides, deparaffinized and rehydration followed by antigen retrieval by incubating samples in citrate buffer (0.01 M, pH 6:0) and heating. Sections were then blocked using BlockAid (ThermoFisher) to preclude non-specific binding. For IHC experiments, endogenous peroxidase activity was quenched by incubating with 3% peroxide for ten minutes. Samples were then incubated in a humidified chamber at 4 C overnight with the primary antibodies as follows: Cleaved caspase-3 (Catalog #: CST9661, Clone: Asp175, Host: Rabbit, Company: Cell Signaling); Ly6g/Gr1 (ab25377, RB6-8C5, Rat, Abcam); CD31 (CST77699, D8V9E, Rabbit, Cell Signaling), and phosphoSTAT3 (CST9145, D3A7, Rabbit, Cell Signaling). The following day, slides were washed and developed using VECTASTAIN R Elite ABC-HRP based kit (Vector) as per the manufacturer's protocol with diaminobenzidine (DAB) as the chromogen. Tissue sections were counterstained with Mayer's hematoxylin, mounted, and imaged.

For immunofluorescence staining (IF), samples were incubated in a humidified chamber at 4 °C overnight with the primary antibodies as follows: Podoplanin (26981, LpMab-12, Mouse, Cell Signaling) and Cxcl1 (ab86436, rabbit, Abcam). Primary antibodies were detected using species-specific Alexa Fluor 594 and/or Alexa Fluor 488 (ThermoFisher) secondary antibodies incubated on sections for one hour at room temperature. Nuclear staining was performed using Hoechst 33342 dye (ThermoFisher).

Quantitative histological analysis was performed by sampling multiple random, non-overlapping 20 x fields in each tissue section (n=5 mice per group) and quantified using ImageJ Fiji software (NIH) to measure the number of positive cells per field, as described previously (*Park et al., 2020*).

## Statistical analysis

Descriptive statistics were calculated using Prism 9.0 (GraphPad, La Jolla, CA). Results are shown as mean ± SEM. Multiple comparisons were performed using one-way ANOVA followed by Tukey's multiple comparisons test. The paired two-sided Student's t-test was used for two-group comparison. An α-cutoff≤0.05 was used to define statistical significance.

## B. Supporting Tables

**Appendix 1—table 1.** Clinicopathological characteristics of study-eligible patients with localized pancreatic ductal adenocarcinoma who received neoadjuvant chemotherapy (NAC) and underwent curative-intent pancreatectomy (BMI: body mass index; HRD: Homologous Recombination Deficiency; ECOG: Easter Cooperative Oncology Group; CAP: College of American Pathology; G-CSF: Granulocyte-colony stimulating factor).

| Variable | All patients (n=94) | Complete/ partial response to NAC (n=66) | Absent/poor response to NAC (n=28) | P-value |
|---|---|---|---|---|
| **Age** (mean ± SD) | 67.3±10.3 | 67.5±10.9 | 66.9±8.9 | 0.78 |
| **Female gender**, n (%) | 58 (61.7 %) | 41 (62.1%) | 17 (60.7%) | 0.89 |
| **Diagnosis BMI,** (mean ± SD) | 26.7±4.9 | 26.8±5 | 26.4±4.6 | 0.68 |
| **Hispanic Ethnicity,** n (%) | 43 (45.8%) | 30 (45.5%) | 13 (46.4%) | 0.64 |
| **Germline HRD mutation,** n (%) | | | | |
| Germline HRD mutation | 6 (6.3%) | 6 (9.1%) | 0 (0%) | |
| No germline HRD mutation | 88 (93.7%) | 60 (91.9%) | 28 (100%) | 0.17 |
| **Pre-chemotherapy Absolute Blood Counts** | | | | |
| Total Leukocyte count ($10^3$ /µl) | 7.2±1.6 | 7.2±1.8 | 7.25±1.5 | 0.92 |
| Neutrophil count (%) | 65.3±7.3 | 63.7±6.9 | 69.7±4.1 | **<0.001** |
| Lymphocyte count (%) | 22.3±7.0 | 24.8±6.7 | 17.6±3.5 | **<0.001** |
| Platelet count ($10^3$ /µl) | 243±71.5 | 246±69.5 | 234±77.3 | 0.59 |
| **Pre-chemotherapy Neutrophil/ Lymphocyte Ratio** (median ± SD) | 3.0±1.3 | 2.5±1.0 | 3.9±1.2 | **<0.001** |
| **Diagnosis CA 19–9 levels** (median ± SD) | 181±1,823 | 147±923 | 202±2,944 | 0.50 |
| **Pre-Surgery Absolute Blood Counts** | | | | |
| Total Leukocyte count ($10^3$ /ul) | 7±1.92 | 6.1±1.72 | 8.5±1.95 | **0.01** |
| Neutrophil count (%) | 63.2±9.5 | 60±7.8 | 71±6.25 | **<0.001** |
| Lymphocyte count (%) | 23.6±7.7 | 26.5±6.4 | 16.7±4.7 | **<0.001** |
| Platelet count (* $10^3$ /ul) | 198±74.1 | 203±57.8 | 181±104 | **0.03** |
| **Pre-Surgery Neutrophil/ Lymphocyte Ratio** (median ± SD) | 2.6±1.57 | 2.3±0.78 | 4.2±1.68 | **<0.001** |
| **ΔNLR (=Pre-Surgery-Pre-Chemo NLR)** (median ± SD) | - 0.1±1.25 | - 0.38±1.1 | 0.21±1.5 | **0.01** |
| **ECOG Status,** n (%) | | | | |
| 0 | 36 (38.3%) | 27 (40.9%) | 9 (32.1%) | |
| 1 | 49 (52.1%) | 34 (51.5%) | 15 (53.6%) | |
| 2 | 9 (9.6%) | 5 (7.6%) | 4 (14.3%) | 0.51 |
| **Tumor location** | | | | |
| Head | 76 (80.8%) | 53 (80.3%) | 23 (82.1%) | |
| Body | 9 (9.6%) | 6 (9.1%) | 3 (10.7%) | |
| Tail | 9 (9.6%) | 7 (10.6%) | 2 (7.2%) | 0.96 |
| **Resectability status** | | | | |
| Resectable | 21 (22.3%) | 15 (22.7%) | 6 (21.4%) | |
| Borderline resectable | 50 (53.2%) | 36 (54.6%) | 14 (50.0%) | |
| Locally advanced | 23 (24.5%) | 15 (22.7%) | 8 (28.6%) | 0.83 |

*Appendix 1—table 1 Continued on next page*

*Appendix 1—table 1 Continued*

| Variable | All patients (n=94) | Complete/ partial response to NAC (n=66) | Absent/poor response to NAC (n=28) | P-value |
|---|---|---|---|---|
| Radiographic tumor size (median ± SD) | 30±14.2 | 30±10.7 | 28.5±20.2 | 0.19 |
| Neoadjuvant Chemotherapy | | | | |
| Gemcitabine/Abraxane | 35 (37.2%) | 26 (39.4%) | 9 (32.1%) | |
| FOLFIRINOX | 49 (52.2%) | 33 (50.0%) | 16 (57.1%) | |
| Both | 10 (10.6%) | 7 (10.6%) | 3 (10.8%) | 0.79 |
| Duration of NAC (months)* | 4±2.3 | 4±2.2 | 4±2.6 | 0.22 |
| Use of G-CSF during NAC | 91 (96.8%) | 64 (97.0%) | 27 (96.4%) | 1.00 |
| Neoadjuvant radiation | 6 (6.4%) | 4 (6.1%) | 2 (7.1%) | 1.00 |
| Histology grade† | | | | |
| Well Differentiated | 2 (2.1%) | 2 (3%) | 0 (0%) | |
| Moderately differentiated | 59 (62.8%) | 49 (74.2%) | 10 (35.7%) | |
| Poorly differentiated | 27 (28.7%) | 11 (16.7%) | 16 (57.1%) | 0.002 |
| pT classification | | | | |
| T1 | 27 (28.7%) | 27 (40.9%) | 0 (0%) | |
| T2 | 33 (35.1%) | 22 (33.3%) | 11 (39.3%) | |
| T3 | 28 (29.8%) | 14 (21.3%) | 14 (50.0%) | |
| T4 | 6 (6.4%) | 3 (4.5%) | 3 (10.7%) | <0.001 |
| pN classification | | | | |
| Positive | 52 (55.3%) | 32 (48.5%) | 20 (71.4%) | |
| Negative | 42 (44.7%) | 34 (51.5%) | 8 (28.6%) | 0.04 |
| Pathological Stage | | | | |
| IA | 17 (18.1%) | 17 (25.7%) | 0 (0%) | |
| IB | 12 (12.8%) | 9 (13.6%) | 3 (10.7%) | |
| IIA | 10 (10.6%) | 6 (9.1%) | 4 (14.3%) | |
| IIB | 47 (50%) | 30 (45.5%) | 17 (60.7%) | |
| III | 7 (7.4%) | 4 (6.1%) | 3 (10.7%) | |
| IV | 1 (1.1%) | 0 (0%) | 1 (3.6%) | 0.04 |
| Neoadjuvant therapy response (CAP grading)‡ | | | | |
| Grade 0 | 0 (0%) | - | | |
| Grade 1 | 12 (12.8%) | 12 (18.2%) | | |
| Grade 2 | 54 (57.4%) | 54 (81.8%) | | |
| Grade 3 | 28 (29.8%) | - | 28 (100%) | N/A |
| R0 resection margin | | | | |
| Yes | 78 (83%) | 61 (92.4%) | 17 (60.7%) | |
| No (R1 resection) | 16 (17%) | 5 (7.6%) | 11 (39.3%) | <0.001 |
| Adjuvant therapy, n (%) | 58 (61.7%) | 43 (65.2%) | 15 (53.6%) | 0.29 |
| Local Recurrence | | | | |
| Yes | 19 (20.2%) | 10 (15.2%) | 9 (32.1%) | |
| No | 75 (79.8 %) | 56 (84.8%) | 19 (67.8%) | 0.06 |
| Distant Recurrence | | | | |
| Yes | 55 (58.5%) | 34 (51.5%) | 21 (75%) | |
| No | 39 (41.5%) | 32 (48.5%) | 7 (25%) | 0.03 |

*Due to variation in dose scheduling between FOLFIRINOX and gemcitabine/abraxane, duration of NAC is reported in months (vs. number of cycles)

†Grade information missing in 6 patients

‡College of American Pathologist (CAP) grading: Grade 0, no viable residual tumor (pathologic complete response); Grade 1, marked response (minimal residual cancer with single cells or small groups of cancer cells); Grade 2, partial response (residual cancer with evident tumor regression, but more than single cells or rare small groups of cancer cells); and Grade 3, poor or no response (extensive residual cancer with no evident tumor regression)

**Appendix 1—table 2.** Salient clinical characteristics and single-cell image segmentation details from imaging mass cytometry experiments comparing tissue-level neutrophil-to-lymphocyte ratio (NLR)

and stromal α-SMA pixel intensity in pre-chemotherapy tissue sections from localized pancreatic ductal adenocarcinoma (PDAC) patients who demonstrated either partial/complete or poor/absent pathologic response to neoadjuvant chemotherapy.

| Pt # | Primary tumor | NAC regimen | Duration of NAC (mo) | Neoadjuvant radiation | Pathologic response | # total single cells in IMC slide | #CD11b+CD15+ neutrophils | #CD3+CD8+ T cells | NLR (norm. to 5000 cells) |
|---|---|---|---|---|---|---|---|---|---|
| 1 | Borderline Resectable | FFX | 4 | No | Partial (CAP 2) | 4326 | 325 | 29 | 11.2 |
| 2 | Borderline Resectable | FFX | 4.5 | No | Partial (CAP 2) | 3256 | 256 | 32 | 7.9 |
| 3 | Locally Advanced* | GNP | 6 | No | Near-complete (CAP 1) | 9421 | 472 | 142 | 3.3 |
| 4 | Borderline Resectable | GNP | 5 | No | Poor/Absent (CAP 3) | 8621 | 951 | 61 | 15.6 |
| 5 | Locally Advanced* | FFX | 6 | No | Poor/Absent (CAP 3) | 6011 | 1105 | 59 | 18.7 |
| 6 | Resectable | FFX +GNP | 6 | No | Poor/Absent (CAP 3) | 4712 | 523 | 40 | 13.1 |

*Representative tissue and image segmentation maps depicted in **Figure 4**

**Appendix 1—table 3.** Predictors of partial/complete pathologic response following neoadjuvant chemotherapy in resected patients with localized pancreatic ductal adenocarcinoma using multivariable logistic regression.

| Variable | OR (95% CI) | P-value |
|---|---|---|
| **Age** | 1.02 (0.92, 1.09) | 0.68 |
| **Gender** | | |
| Female | *Ref* | - |
| Male | 2.86 (0.45, 26.2) | 0.34 |
| **Diagnosis BMI** | 0.88 (0.74, 1.08) | 0.21 |
| **CA 19–9 dynamics** | | |
| Any increase | *Ref* | - |
| Any decrease | 1.82 (0.001, 3.74) | **0.05** |
| **Resectability Status** | | |
| Borderline | *Ref* | - |
| Locally advanced | 2.72 (0.22, 33.2) | 0.43 |
| Resectable | 0.64 (0.06, 7.44) | 0.72 |
| **Radiographic tumor size** | 0.98 (0.91, 1.05) | 0.54 |
| **NAC duration (months)** | 1.09 (0.68, 1.75) | 0.73 |
| **Absolute pre-chemotherapy aNLR** | | |
| Low | *Ref* | - |
| High | 0.02 (0.003, 0.15) | **<0.001** |
| **ΔNLR** | | |
| Low | *Ref* | - |
| High | 0.06 (0.01, 0.33) | **0.002** |

## C.Supporting Figures

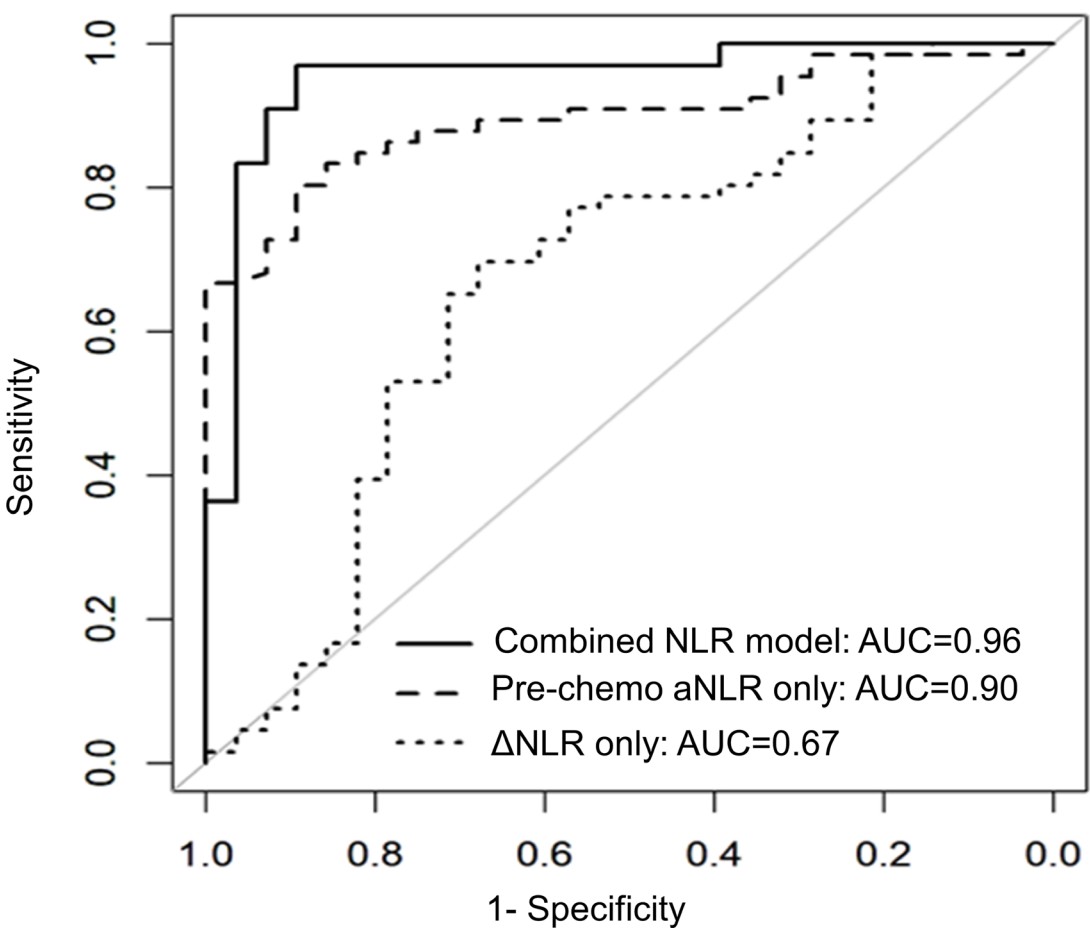

**Appendix 1—figure 1.** NLR dynamics during neoadjuvant chemotherapy (NAC) are associated with pathologic response in patients undergoing resection for pancreatic cancer. Area under the receiver operating characteristic curve (AUC) statistics estimating the predictive capacity of three biomarker models (pre-chemotherapy aNLR only, ΔNLR only, combined model aNLR + ΔNLR) for pathologic response, internally validated with bootstrap logistic regression.

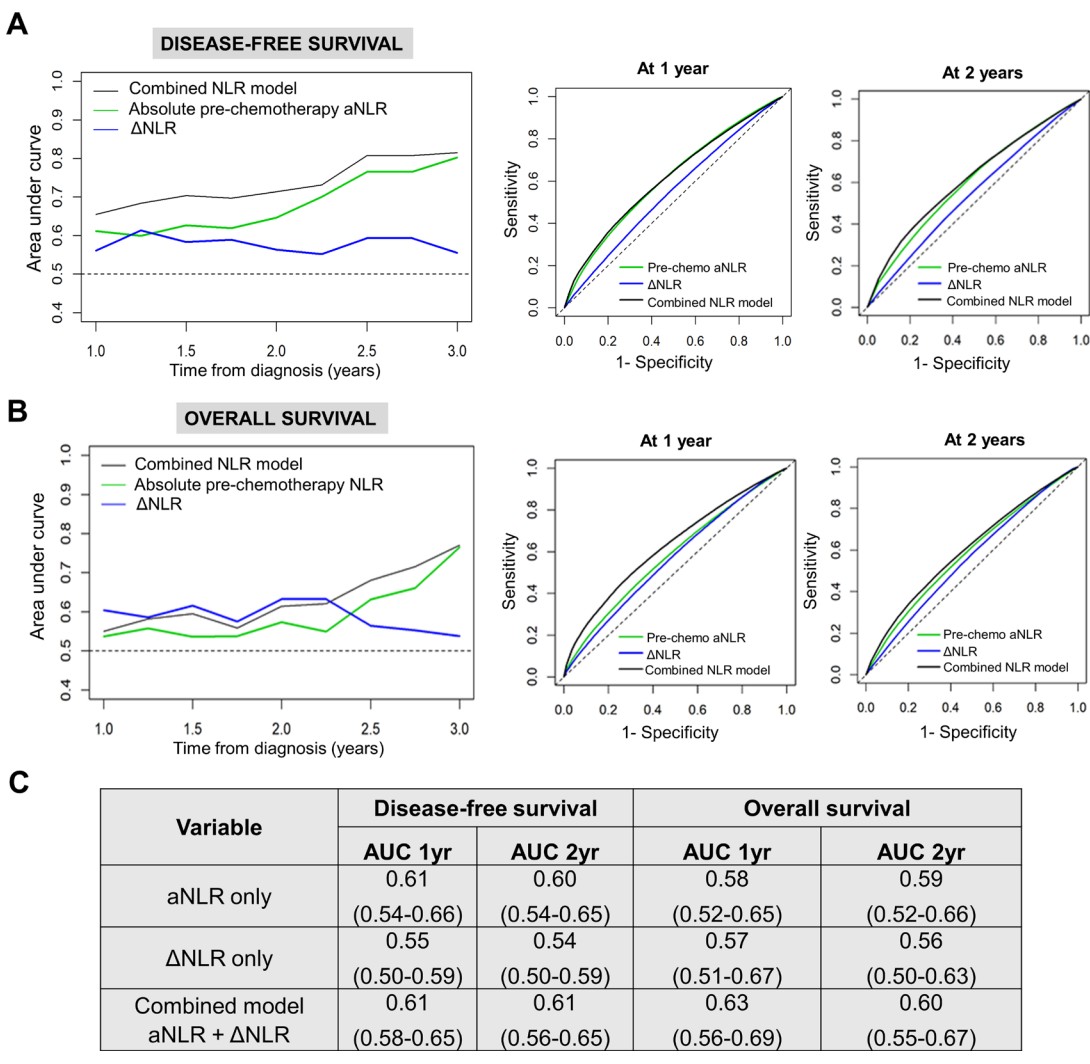

**Appendix 1—figure 2.** NLR dynamics during neoadjuvant chemotherapy (NAC) are associated with survival in patients undergoing resection for pancreatic cancer. Time-dependent AUC analysis with internal bootstrap validation examining the three biomarker models for (**A**) disease-free survival (DFS; *left*) and (**B**) overall survival (OS; *left*) for years 1–3. Corresponding time-specific receiver operating characteristic curves for 1 year and 2 years for both DFS and OS are shown (*right*); (**C**) Table showing reported mean AUC along with 95% confidence intervals for the three predictive biomarker models at 1 year and 2 years for DFS and OS.

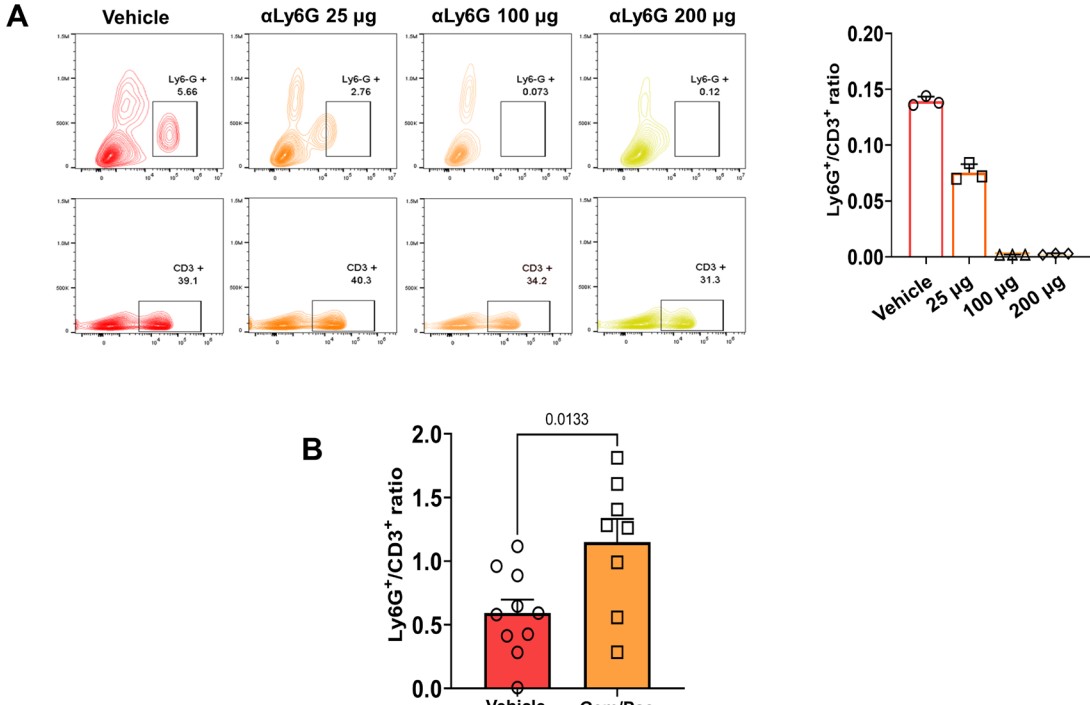

**Appendix 1—figure 3.** Titration of non-depleting NLR attenuating anti-Ly6G antibody dosing in a preclinical model of PDAC. (**A**) Representative flow cytometry contour plots depicting circulating splenocyte-derived Ly6G$^+$ (*top*) and CD3$^+$ (*bottom*) cell populations 2 weeks following anti-Ly6G dose titration experiment to identify NLR attenuating—but not depleting—dose. Histograms represent ratio of Ly6G$^+$:CD3$^+$ cells across different treatment arms (n=3 mice each). Based on these studies, we selected 25 µg dose because it reduced NLR by 50%; (**B**) Histogram representing Ly6G$^+$:CD3$^+$ ratio in vehicle treatment vs. gemcitabine/paclitaxel treatment to demonstrate the effect on circulating NLR with chemotherapy treatment in our murine model (n=8–10 mice per group).

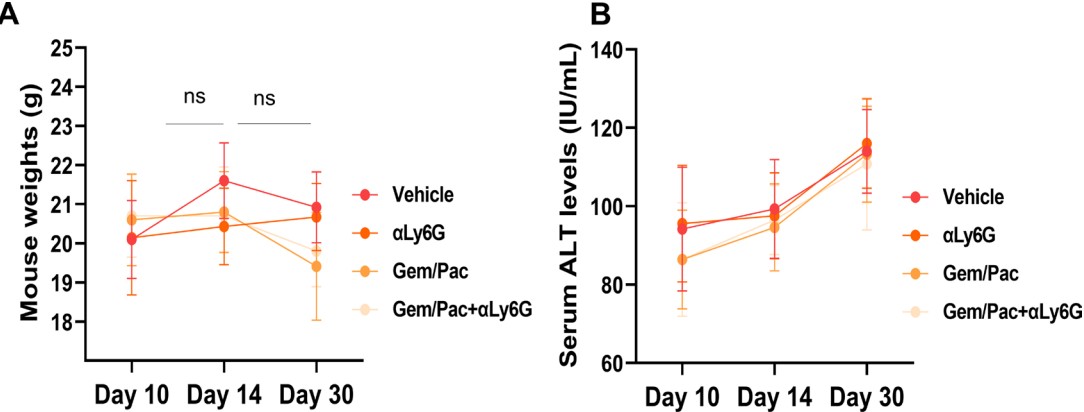

**Appendix 1—figure 4.** NLR attenuation with or without systemic chemotherapy in preclinical models of PDAC does not increase treatment-related toxicity. (**A**) Mean (± standard deviation) body weights, and (**B**) mean ± SD of alanine transferase (ALT) levels from blood of mice in each cohort (n=8–10/group) graphed at 3 time points based on treatment initiation and sacrifice, to assess treatment-related toxicity; ns: not significant.

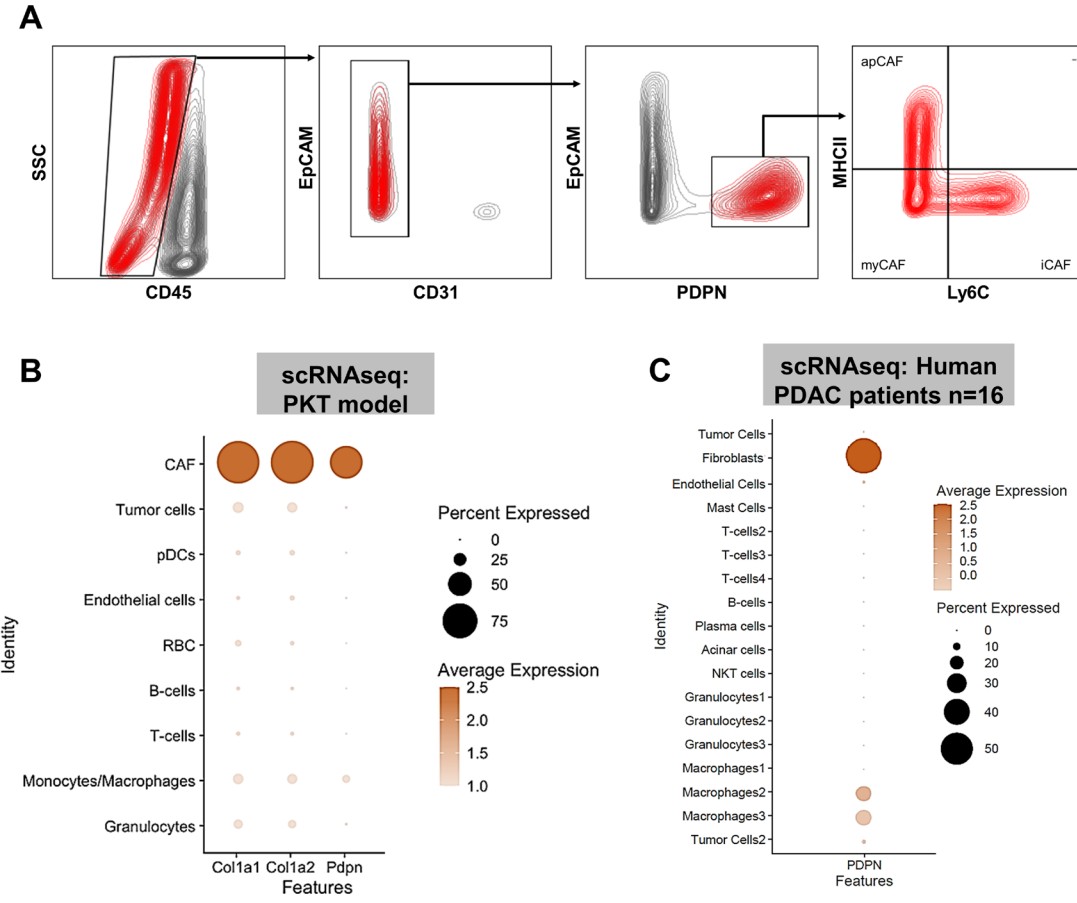

**Appendix 1—figure 5.** PDPN is a marker of pancreatic tumor-associated cancer associated fibroblasts (CAF). (**A**) Gating strategy for flow cytometric analysis of PDPN+ CAF populations after exclusion of CD45+, EpCAM+, and CD31+ cells; (**B**) Dot plot from single cell RNA sequencing (scRNAseq) dataset [reference 13] showing near-exclusive expression of *Pdpn* in CAF subcluster nominated by *Col1a1* and *Col1a2* expression; (**C**) scRNAseq data from human PDAC patients [reference 14] showing near-exclusive expression of *PDPN* in tumor-associated fibroblast clusters. In (**B**) and (**C**), expression density and percent expression in respective sub-cluster is indicated in adjoining legends.

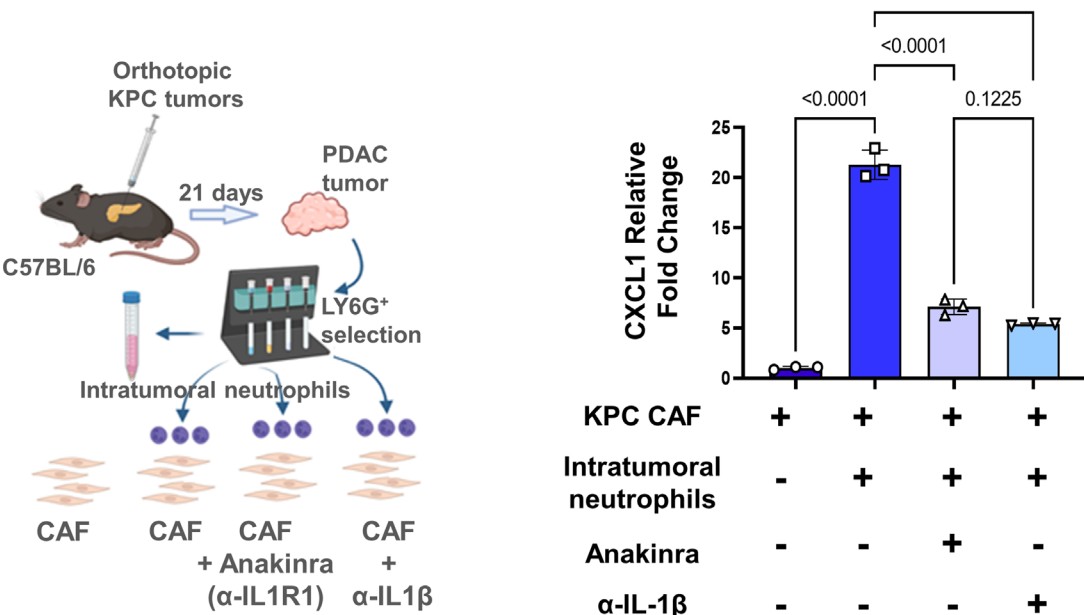

**Appendix 1—figure 6.** Neutrophil-derived IL-1β is a novel mediator of inflammatory CAF polarization in pancreatic cancer. Schematic of experimental design illustrating ex vivo co-culture of KPC CAFs with intratumoral column-sorted Ly6G⁺F4/80 cells from whole pancreata of KPC orthotopic mice, with or without pre-treatment of CAFs anakinra (α-IL1R1 antibody) or pre-treatment of neutrophils with α-IL-1β neutralizing antibody (*left*). qPCR analysis representing relative fold change in *Cxcl1* gene expression comparing CAFs alone with CAFs co-cultured with intra-tumoral neutrophils with or without anakinra or α-IL-1β antibody pre-treatment (*right*). Results show mean ± SEM of three biologic replicates; *, $p<0.05$; **, $p<0.01$; ***, $p<0.001$

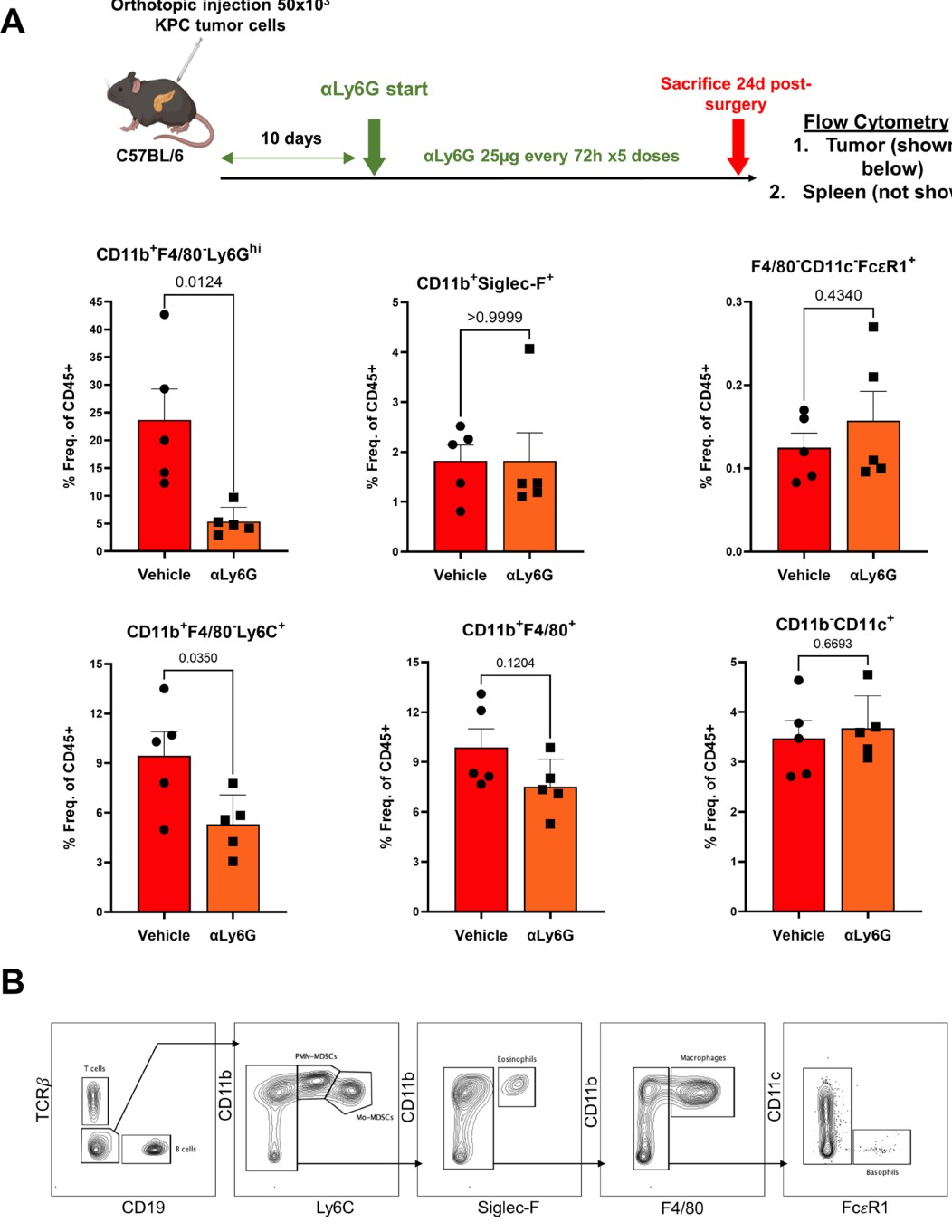

**Appendix 1—figure 7.** Anti-Ly6G antibody specifically targets PDAC-associated neutrophils, but not other granulocytic populations. (**A**) Schematic of experimental design showing timing/dosing of anti-Ly6G treatment in orthotopic KPC tumor-bearing mice (*top*), with adjacent histograms showing (% of parent CD45+ cells) relative proportions of intratumoral neutrophils/PMN-MDSCs (CD11+F4/80-Ly6Ghi), eosinophils (CD11b+F4/80-Siglec-F+), basophils (F4/80-CD11c-FcεR1+), monocytic-MDSCs (CD11b+F/480-Ly6C+), macrophages (CD11b+F4/80+), and dendritic cells (CD11b-CD11c+); (**B**) Gating strategy for flow cytometric analysis shown in (**A**).

