## [Editor Report]

Iago De Castro et al. is a fundamental new study that conveys to readers that neutrophil-to-lymphocyte ratio dynamics could predict pancreatic cancer pathologic response to neoadjuvant therapy. The study is compelling in that it specifically provides means to determine the effect that front-line neoadjuvant therapy could have on the function of key microenvironmental cells (e.g., T Cell and Cancer-associated fibroblast) if combined with an anti-Ly6G treatment.

---

## [Decision Letter]

**Decision letter after peer review:**

Thank you for submitting your article "Neutrophil-Mediated Stromal-Tumor IL-6/STAT-3 Signaling Underlies the Association between Neutrophil-to-Lymphocyte Ratio Dynamics and Chemotherapy Response in Localized Pancreatic Cancer: A Hybrid Clinical-Preclinical Study" for consideration by *eLife*. Your article has been reviewed by 2 peer reviewers, and the evaluation has been overseen by myself, acting as the Reviewing Editor, and by Wafik El-Deiry as the Senior Editor. The following individual involved in the review of your submission has agreed to reveal their identity: Mara Sherman (Reviewer #2).

We have discussed the reviews with one another, and I have drafted this to help you prepare a revised submission.

We believe your study should be revised prior to its acceptance in order to address the following four main points:

1. Relevance to the human condition.

2. Stronger link to CAF transitions.

3. Considering/discussing previous studies related to yours.

4. Strengthening the anti-Ly6G specificity proof.

*Reviewer #1 (Recommendations for the authors):*

1. My concern is the relevance of CAFs to the correlations observed in the human cohort. The authors should strengthen this part of the study to support the claim that CAF subpopulations transition also in the human disease. Do patients with low neutrophil infiltration have less iCAFs and a better prognosis? The authors should show this by immunostaining of human patient samples for which NLR analysis was done with CAF markers.

2. Following up on the previous comment, the data for CAF transitions is based on FACS with negative selection for immune, endothelial and epithelial markers and positive selection for PDPN. While these are most likely CAFs, PDPN is not an exclusive marker for CAFs. Since this data is not shown in the human cohort, the authors should perform immunofluorescent staining to show, in situ, that the cells affected by these treatments are indeed CAFs.

3. Statistical tests: Some of the differences shown do not seem to be significant. For example, in Figure 2A, C – did the authors correct for multiple comparisons? This should be detailed in the legends and methods.

4. Figure 3: FACS results in viSNE maps are not clear. The scale is unclear and it seems that anti-Ly6G+ expresses Ly6G+. The T cell panel also doesn't appear to indicate a decrease in PD-1. The authors should also provide histograms or dot plots.

5. Figure 4: Anti-Ly6G treatment resulted in a significant reduction of IL-1 secretion, however, the authors did not show any effect of gemcitabine/paclitaxel treatment, since this treatment could increase IL-1.

*Reviewer #2 (Recommendations for the authors):*

The prognostic significance of NLR for response to neoadjuvant chemotherapy among PDAC patients has been assessed in prior studies not presently cited here, including PMID: 33517697 and PMID: 31549202. The former (Strong et al., Am Surg, 2021) addresses NLR dynamics during neoadjuvant treatment, and in this study, the authors concluded that change in NLR does not predict pathologic response or survival among resectable PDAC patients. It is important for de Castro Silva et al. to discuss this study and provide a rationale that may underlie these distinct conclusions. In addition, while the preclinical results using anti-Ly6G in vivo are exciting, Ly6G is also expressed by granulocytes and some monocytes in addition to neutrophils. The authors should either state this caveat with respect to the lack of specificity for neutrophils in the manuscript text or inhibit neutrophils in vivo using an orthogonal approach to strengthen their claims with respect to neutrophil function. Finally, in light of the heterogeneity within the myeloid compartment observed across PDAC mouse models as well as patients, it would be informative to test the therapeutic potential of anti-Ly6G plus chemotherapy in an independent mouse model and report a limited number of key results (i.e., tumor growth and metastatic spread).

---

## [Author Response]

We believe your study should be revised prior to its acceptance in order to address the following four main points:1. Relevance to the human condition.

We have added a substantial series of experiments in human PDAC tumors, utilizing single-cell imaging mass cytometry and immunofluorescence to corroborate the novel relationship between tissue-level NLR, stromal inflammation/density, and chemotherapy response in human PDAC. Please see response to Reviewer 1 comment #1.

2. Stronger link to CAF transitions.

We have provided new data in PKT mice and human tumor sections showing co-immunofluorescence staining of stromal PDPN and CXCL1—reflective of iCAF populations—to strengthen data previously shown by flow cytometry in the original manuscript. We have also shown single cell RNA sequencing data from our recent publication (Datta et al., Gastroenterology 2022) to reinforce the relevance of CD45^neg^CD31^neg^PDPN^pos^ as a bona fide CAF marker. Please see response to Reviewer 1 comment #2 and Reviewer 2 comment #3.

3. Considering/discussing previous studies related to yours.

We have included indicated references in the revised Introduction and Discussion.

4. Strengthening the anti-Ly6G specificity proof.

We have performed new experiments demonstrating the specificity of NLR-attenuating anti-Ly6G antibody for neutrophils, but not eosinophil or basophil populations in vivo. Please see response to Reviewer #2 comment #2.*Reviewer #1 (Recommendations for the authors):*

1. My concern is the relevance of CAFs to the correlations observed in the human cohort. The authors should strengthen this part of the study to support the claim that CAF subpopulations transition also in the human disease. Do patients with low neutrophil infiltration have less iCAFs and a better prognosis? The authors should show this by immunostaining of human patient samples for which NLR analysis was done with CAF markers.

We thank the reviewer for this important suggestion. We have now made substantial additions to the revised manuscript to address this deficit. We identified 6 human PDAC tumors with pre-chemotherapy tissue, and stratified these based on ultimate post-chemotherapy pathologic response into partial/complete vs. poor/absent. We then used a novel Hyperion imaging mass cytometry (IMC) platform to perform single-cell image segmentation and examine the relationship between tissue-level NLR (CD11b^+^CD15^+^ neutrophil:CD3^+^CD8^+^ T-cell ratios), chemotherapy response, and stromal density. These clinical and relevant IMC data can be found in the new Appendix-Table 2.

To further examine differences in stromal inflammation in these clinically annotated human PDAC samples, we performed co-IF for PDPN (see response to comment #2) and CXCL1 (iCAF marker). Results are highly supportive of our main conclusion, and validate recent findings reported from the WashU group suggesting that iCAF polarization in the PDAC tumor microenvironment is associated with chemotherapy resistance (Zhou et al., Nat Genet 2022).

Our findings are described in the revised Results under a new section (pg. 12):

“Reduced tissue-level NLR correlates with chemotherapy response, CAF density, and stromal inflammation at single-cell resolution in human PDAC. To examine the association between tissue-level NLR, stromal density/inflammation, and chemotherapy response (partial/complete [n=3], poor/absent [n=3]) in human PDAC tumors at single-cell resolution, pathologist-selected regions of interest (ROI) from each tumor section probed with metal ion-conjugated antibodies for pancytokeratin (PanCK:epithelial), α-smooth muscle actin (α-SMA:fibroblast), CD11b and CD15 (neutrophil), and CD3 and CD8 (T-cell) were laser-ablated, and atomized ions were acquired using time-of-flight mass cytometry (cyTOF) (Figure 4C). Image segmentation and quantification revealed significantly higher ratio of CD11b^+^CD15^+^ to CD3^+^CD8^+^ cells (NLR; normalized to 5000 total single cells) in pre-treatment tumors from PDAC patients who demonstrated poor/absent pathologic response compared with partial/complete response (15.8±2.8 vs. 7.4±3.9; P=0.039) following neoadjuvant chemotherapy (Figure 4D). Interestingly, increased NLR in patients with poor/absent pathologic response correlated with significantly higher mean intensity of α-SMA expression (41.9±26.6 vs. 18.4±16.6 pixels/cell; P<0.001) in—but not absolute density of—cancer associated fibroblasts in tumor ROIs (Figure 4D), as well as relative abundance of co-expressed PDPN^+^CXCL1^+^ iCAF populations in corresponding tumor sections (29.7±8.8% vs. 18.4±7.4% tumor area; P<0.001; Figure 4E).

2. Following up on the previous comment, the data for CAF transitions is based on FACS with negative selection for immune, endothelial and epithelial markers and positive selection for PDPN. While these are most likely CAFs, PDPN is not an exclusive marker for CAFs. Since this data is not shown in the human cohort, the authors should perform immunofluorescent staining to show, in situ, that the cells affected by these treatments are indeed CAFs.

(1) PDPN has been used as a pan-CAF marker in multiple high-profile PDAC studies. A few examples with relevant points are presented here—PMID 31699795: PDPN is expressed in fibroblasts and CAFs; CD31^+^ stromal cells were predominantly PDPN^−^ blood endothelial cells; PMID 33495315: Used PDPN to define CAFs both via flow cytometry and IHC; PMID 31197017: Defined PDPN as a pan-CAF marker; PMID 30366930: PDPN used to sort and isolate CAFs; PMID 30575030: PDPN used to identify CAFs in human samples, including IHC.

(2) We have also invoked some of these studies in our revised manuscript (pg. 12). We now show single-cell RNA sequencing in both the spontaneous PKT PDAC model (Datta J et al., Gastroenterology 2022), as well as human PDAC patients (Steele et al., Nat Cancer 2020) demonstrating near-exclusive expression of PDPN in fibroblast populations in the murine and human PDAC tumor microenvironments. These data are now invoked in the revised Results (pg. 12): “Furthermore, leveraging the near-exclusive expression of PDPN/Pdpn in human and murine PDAC-associated CAFs via scRNAseq (Appendix-Figure 5B&C)…”

(3) Endothelial cells are the other dominant cell subset in which PDPN is expressed. For this reason, we gated out CD31^+^ cells prior to examining Ly6C and MHC-II markers in PDPN^+^ cells. This gating strategy has been validated previously (Biffi et al., Cancer Discovery 2019) and we have now shown our strategy in Appendix-Figure 5C.

(4) We have now performed immunofluorescence co-staining of PDPN and CXCL1 in both PKT genetic murine tumors, as well as in human PDAC sections (see response to previous comment) as an orthogonal approach to show the relevance of PDPN as a CAF marker and PDPN^+^CXCL1^+^ fibroblasts as representative of inflammatory CAFs. The data from PKT mice are described in the revised Results (pg. 12): “We observed significant reduction in co-expressing PDPN^+^CXCL1^+^ stromal cells—presumed iCAFs—in tumors from PKT genetically engineered mice treated with gemcitabine/paclitaxel+anti-Ly6G compared with gemcitabine/paclitaxel alone (P=0.02; Figure 4B), validating findings from the KPC orthotopic model.”.

3. Statistical tests: Some of the differences shown do not seem to be significant. For example, in Figure 2A, C – did the authors correct for multiple comparisons? This should be detailed in the legends and methods.

We apologize for the oversight. Yes, the comparisons shown in Figures 2A-C were performed using ANOVA with Tukey’s multiple-comparison post-hoc testing. This is now detailed in the revised Appendix (Supplementary Methods) section: “Results are shown as mean ± SEM. Multiple comparisons were performed using one-way ANOVA followed by Tukey’s multiple comparisons test.” We have also updated the Figure 2, 4, 5 legends to reflect these edits (pg. 22-23, 27, 28).

4. Figure 3: FACS results in viSNE maps are not clear. The scale is unclear and it seems that anti-Ly6G+ expresses Ly6G+. The T cell panel also doesn't appear to indicate a decrease in PD-1. The authors should also provide histograms or dot plots.

We have improved the readability of the viSNE maps (see below). Regarding scant Ly6G expression in anti-Ly6G treated mice, we gently remind the reviewer that our intention was not to deplete but attenuate NLR (please see Appendix-Figure 3A) to mimic NLR attenuation in PDAC patients. We have reinforced this in the revised Supplementary Methods in Appendix: “Our specific intention was to attenuate, but not deplete, systemic neutrophils to simulate the endogenous decline in ΔNLR during neoadjuvant chemotherapy”, as well as revised Results (pg. 11): “In tumor-bearing animals, NLR attenuation significantly reduced—but did not abolish—circulating Ly6G^+^Ly6C^dim^F4/80^-^ neutrophilic cells (Figure 3A).” We also provide adjoining histograms for these comparisons in Figure 3A.

As we and multiple groups have done previously, we use PD-1 as a marker of T-cell antigen experience (and not exhaustion per se, which has a more complex expression profile). We actually observed a significant increase in mean fluorescence intensity (MFI) of PD-1 in tumor-infiltrating CD8^+^ T-cells—but not absolute number of PD-1^+^CD8^+^ T-cells—in chemotherapy+anti-Ly6G groups compared with either treatment alone. To clarify these comparisons between groups, we have removed prior Appendix-Figure 5 and have provided these histograms adjacent to the viSNE maps in Figure 3C.

5. Figure 4: Anti-Ly6G treatment resulted in a significant reduction of IL-1 secretion, however, the authors did not show any effect of gemcitabine/paclitaxel treatment, since this treatment could increase IL-1.

We thank the reviewer for this insightful comment. We have now performed and shown these data in the revised manuscript. As the reviewer predicted, gemcitabine/paclitaxel chemotherapy treatment modestly (but not significantly) increases IL-1β, but combination anti-Ly6G+chemotherapy treatment recapitulates the effect of anti-Ly6G alone by significantly reducing IL-1β secretion compared with chemotherapy alone. These findings mirror the reduction in intratumoral Ly6G/Gr1^+^ neutrophilic cells observed in both anti-Ly6G treated cohorts. To address the reviewer comment, these data are now described in revised Results (pg. 13): “Systemic NLR attenuation with anti-Ly6G treatment—with or without chemotherapy—resulted in significant diminution of IL-1β secretion in tumor lysates compared with vehicle or chemotherapy treatment in vivo (ANOVA P<0.001; Figure 5E), likely due to its incident reduction in systemic and tumor-infiltrating Ly6G^+^ cells (see Figure 3).”

Reviewer #2 (Recommendations for the authors):The prognostic significance of NLR for response to neoadjuvant chemotherapy among PDAC patients has been assessed in prior studies not presently cited here, including PMID: 33517697 and PMID: 31549202. The former (Strong et al., Am Surg, 2021) addresses NLR dynamics during neoadjuvant treatment, and in this study, the authors concluded that change in NLR does not predict pathologic response or survival among resectable PDAC patients. It is important for de Castro Silva et al. to discuss this study and provide a rationale that may underlie these distinct conclusions.

We thank the reviewer for this important comment. While acknowledging that the Strong et al., study (PMID: 33517697) has a different conclusion than our study, several issues limiting its applicability to our findings should be recognized and discussed:

1) These data are presented from an institution that sequences most patients with localized disease to a surgery-first approach, calling into question a substantial selection bias that compels utilization of neoadjuvant therapy and confounds the study conclusions.

2) Utilization of non-standard pathologic response nomenclature—their study dichotomizes response by < or ≥90% response—which is not standard in the field. Our study uses standard College of American Pathologists (CAP) criteria to categorize pathologic response. It is unknown how these two response metrics compare.

3) There is concerning covariate imbalance in good vs. poor responders in their study, wherein 88% of patients with “good” pathologic response underwent neoadjuvant radiation vs. only 36% of patients with poor response (P<0.001). Moreover, this variable was not accounted for when examining the impact of NLR on pathologic response.

4) Another imbalance was observed in the duration of chemotherapy between the cohorts, with “good” responders receiving median 22 weeks of NAC vs. only 12 weeks in the “poor” responders. Again, this variable is not accounted for in a multivariable analysis. Our analysis, on the other hand, controls for duration of NAC in the multivariable analysis (OR 1.09, 95% CI 0.68-1.75, P=0.73), which is shown in Appendix-Table 3.

5) Of 93 patients included in the study, 14 patients had unknown baseline NLR values, of which 12 were in the poor responder cohort.

6) Finally, despite these major issues, there was a suggestion that “good” responders actually had attenuation of NLR during chemotherapy (median ΔNLR = -0.02) compared with “poor” responders (median ΔNLR = +0.06; Table 2), although this comparison was not statistically significant. These findings are consistent with our present study.

Despite these major issues, we have incorporated this reference in the revised Discussion (pg. 14): “While these novel findings warrant large-scale multi-institutional validation to strengthen and/or reconcile data from heterogeneous PDAC populations [PMID: 26893780, 31549202, 33517697]…”

We apologize for overlooking the Japanese study that the reviewer points out (PMID: 31549202), which supports our overall conclusions. We have invoked this reference in the revised Introduction (pg. 4): “…recent evidence implicates the value of pre-surgery NLR in forecasting recurrence in patients undergoing upfront pancreatectomy [4], as well as pre- and post-treatment NLR in predicting pathologic response following neoadjuvant chemoradiotherapy [5,6].”

In addition, while the preclinical results using anti-Ly6G in vivo are exciting, Ly6G is also expressed by granulocytes and some monocytes in addition to neutrophils. The authors should either state this caveat with respect to the lack of specificity for neutrophils in the manuscript text or inhibit neutrophils in vivo using an orthogonal approach to strengthen their claims with respect to neutrophil function.

We thank the reviewer for this insightful comment. We now provide evidence that NLR-attenuating doses of anti-Ly6G Ab specifically disrupt intratumoral neutrophils/neutrophilic MDSCs and to a lesser extent monocytes/monocytic MDSCs. More importantly, to address the reviewer’s comment, we do not observe any differences in other granulocytic populations between vehicle- and anti-Ly6G Ab-treated cohorts, namely comparing tumoral (and circulating) CD11b^+^F4/80^-^Siglec-F^+^ eosinophils and F4/80^-^CD11c^-^FcεR1^+^ basophils. These data have been provided in the Supplementary Methods section in Appendix: “The anti-Ly6G clone 1A8 neutralizing antibody construct is a rat IgG2a that induces a Fc-dependent opsonization and phagocytosis of Ly6G^+^ cells. To demonstrate the specificity of this antibody, a separate experiment showed that NLR-attenuating anti-Ly6G treatment in KPC orthotopic tumor-bearing mice specifically reduced splenic (data not shown) and intratumoral CD11b^+^F4/80^-^Ly6G^+^ neutrophils, but not other granulocytic populations—namely CD11b^+^F4/80^-^Siglec-F^+^ eosinophils and F4/80^-^CD11c^-^FcεR1^+^ basophils—compared with vehicle treatment (Appendix-Figure 7).”

With respect to an orthogonal approach to constraining neutrophils and its effect on chemosensitivity, data from Linehan and colleagues have revealed that CXCR2 inhibition sensitizes PDAC to FOLFIRINOX chemotherapy. We have invoked these prior data in the revised Discussion (pg. 15) but did not repeat these experiments in the present manuscript: “…or that ablation of CXCR2^+^ tumor-associated neutrophils augments IFN-γ^+^CD8^+^ T-cell infiltration to potentiate FOLFIRINOX responses in PDAC models (Nywening TM et al., Gut 2018).”

Finally, in light of the heterogeneity within the myeloid compartment observed across PDAC mouse models as well as patients, it would be informative to test the therapeutic potential of anti-Ly6G plus chemotherapy in an independent mouse model and report a limited number of key results (i.e., tumor growth and metastatic spread).

We agree with this important comment. To address the reviewer’s suggestion, we performed pharmacologic Ly6G attenuation with gemcitabine+paclitaxel chemotherapy vs. chemotherapy alone in the genetically engineered Ptf1a^cre/+^;LSL-Kras^G12D/+^;Tgfbr2^flox/flox^ (PKT) mouse model. We have published extensively on this model—mice form aggressive tumors by week 4 and die of local (but not metastatic) disease burden by 6.5 weeks (e.g., Datta J et al., Gastroenterology 2022; Nagathihalli et al., Cancer Res 2018). We have detailed this experimental design in Appendix.

We performed an endpoint experiment after 2 weeks of anti-Ly6G treatment + chemotherapy vs. chemotherapy alone vs. vehicle (omitted anti-Ly6G alone for these experiments), and observed significantly reduced tumor burden in cohorts treated with anti-Ly6G+chemo vs. chemo alone. We also observed reduced tumor area upon histologic analysis by a board-certified GI pathologist. These results are now described in the revised Results (pg. 10): “To validate these observations in a spontaneous PDAC mouse model, we treated 4-week old PKT mice with vehicle, gemcitabine/paclitaxel alone, and gemcitabine/paclitaxel plus anti-Ly6G combinations for 2 weeks. In this model as well, NLR attenuation with anti-Ly6G improved chemosensitivity vs. chemotherapy alone as evidenced by decreased primary tumor weights (P=004; Figure 2E) and tumor area by H&E staining (P=0.008; Figure 2F) at endpoint analysis.”

In addition, we validated the reduction in inflammatory CAF populations—characterized by PDPN^+^CXCL1^+^ stromal cells—in the combination anti-Ly6G+gemcitabine/paclitaxel treatment arm compared with chemotherapy alone or vehicle treatment in PKT mice. These data are now described in the revised Results (pg. 12): “Furthermore … we observed significant reduction in co-expressing PDPN^+^CXCL1^+^ inflammatory CAF populations in tumors from PKT genetically engineered mice treated with gemcitabine/paclitaxel+anti-Ly6G compared with gemcitabine/paclitaxel alone (P=0.02; Figure 4B), validating findings from the KPC orthotopic model.”